# Sources, cycling and export of nitrogen on the Greenland Ice
# Sheet
*Wadham[1], J.L., J. Hawkings[1], J. Telling[1], D. Chandler[1], J. Alcock[1], E. O'Donnell[2], P.
Kaur[1], E.A. Bagshaw[1], M. Tranter[1], A. Tedstone[3], P. Nienow[3]
[1] – Bristol Glaciology Centre, School of Geographical Sciences, University of Bristol,
University Road, BS8 1SS, UK
[3] - School of Geography, University of Nottingham, NG7 2RD, UK
[2] - School of Geoscience, University of Edinburgh, Edinburgh, EH8 9XP, UK
Correspondence to: J.L. Wadham (j.l.wadham@bris.ac.uk)

**Abstract**
Fjord and continental shelf environments in the Polar Regions are host to some of the planet's
most productive ecosystems, and support economically important fisheries. Their productivity,
however, is often critically dependent upon nutrient supply from upstream terrestrial
environments delivered via river systems. In glacially-fed coastal ecosystems, riverine nutrients
are largely sourced from melting snow and ice. The largest and most extensive glacially-fed
coastal ecosystem in the Arctic is that bordering the Greenland Ice Sheet. The future primary
productivity of this ecosystem, however, is uncertain. A potential increase in primary
productivity driven by reduced sea ice extent and associated increased light levels may be
curtailed by insufficient nutrient supply, and specifically nitrogen. Research on small valley
glaciers indicates that glaciers are important sources of nitrogen to downstream environments.
However, no data exists from ice sheet systems such as Greenland. Time series of nitrogen
concentrations in runoff are documented from a large Greenland glacier, demonstrating
seasonally elevated fluxes to the ocean. Fluxes are highest in mid-summer, when nitrogen
limitation is commonly reported in coastal waters. It is estimated that approximately half of the
glacially-exported nitrogen is sourced from microbial activity within glacial sediments at the

surface and bed of the ice sheet, doubling nitrogen fluxes in runoff. Summer dissolved inorganic nitrogen fluxes from the Greenland Ice Sheet (30-40 Gg) are a similar order of magnitude to those from a large Arctic river (Holmes et al., 2012). Nitrogen yields from the ice sheet (235 kg TDN km$^{-2}$ a$^{-1}$), however, are approximately double those from Arctic riverine catchments. We assert that this ice sheet nitrogen subsidy to Arctic coastal ecosystems may be important for understanding coastal biodiversity, productivity and fisheries, and should be considered in future biogeochemical modelling studies of coastal marine productivity in the Arctic regions.

## 1. Introduction

The availability of nitrogen widely limits primary productivity in fjord (Rysgaard et al., 1999), coastal (Poulsen and Reuss, 2002;Daly et al., 1999;Nielsen and Hansen, 1999) and open ocean (Smith et al., 1985;Moore et al., 2002) waters bordering the Greenland Ice Sheet (GrIS) in summer. Hence, external sources of nitrogen to these waters, e.g. riverine runoff, may be important in sustaining the productivity of these waters and may alter in a warming climate. These Greenlandic waters are some of the most productive ecosystems in the world, and boast high socio-economic value via fisheries (e.g. shrimp, halibut) (Hamilton et al., 2000). In the North Atlantic, primary productivity also draws down $CO_2$ from the atmosphere and has an important regulatory effect on global climate (Sabine et al., 2004). Warmer ocean temperatures and a lengthened growing season in the Arctic are predicted in future decades. However, increases in marine primary productivity may be capped by intensified summer nitrogen limitation (Vancoppenolle et al., 2013).

The GrIS discharges >1000 km$^3$ of freshwater annually to the Arctic Ocean, Irminger Sea, Labrador and Greenland Seas (Bamber et al., 2012) but has yet to be evaluated as a source of nitrogen to these waters. This freshwater flux is increasing (Bamber et al., 2012), and will continue to do so as rising air and ocean temperatures enhance rates of ice sheet melting and iceberg calving (IPCC, 2007). Greenland ice core data show the ubiquitous presence of low concentrations of dissolved inorganic nitrogen (DIN) in ice and snow, sourced from the atmosphere (Wolff, 2013). Based upon findings from small glacier systems (Hodson et al., 2008;Telling et al., 2011;Boyd et al., 2011), it is plausible that this atmospheric DIN is supplemented by nitrogen cycled into bioavailable forms by glacial biota (Telling et al., 2012;Boyd et al., 2011). While there is a mounting body of literature on nitrogen cycling on

valley glaciers (Telling et al., 2011;Hodson et al., 2008), there is comparatively little data on
nitrogen sources and cycling on the Greenland Ice Sheet, which is likely to be important as a
nutrient source to downstream fjord and marine ecosystems. High reported rates of fjord primary
productivity around the GrIS margin (Jensen et al., 1999) and coastal blooms as late as
July/August (Frajka-Williams and Rhines, 2010;Nielsen and Hansen, 1999) coincident with peak
meltwater fluxes (Bartholomew et al., 2011) suggest that such an evaluation will be a fruitful
exercise.
This manuscript aims to examine the sources, cycling and fluxes of nitrogen and it's
component species in bulk runoff exported from the GrIS during the summer melt season. We
document seasonal time series of nitrogen concentrations, speciation and fluxes associated with
both surface meltwaters and subglacial runoff at the large (600 km$^2$) land-terminating Leverett
Glacier (LG) in SW Greenland during the 2012 melt season. This glacier has a bedrock
(Precambrian gneiss/granite (Kalsbeek, 1982)) that is consistent with large areas of the GrIS, and
covers a large altitudinal range drawing meltwaters from >100 km from the ice margin. Hence,
we assert that it is representative of large sectors of the Greenland margin. This manuscript
builds upon recent work by (Hawkings et al., 2015), who presented a more limited dataset of
dissolved inorganic subglacial nitrogen fluxes from the same catchment. We report data from a
range of contextual field and experimental samples collected from sediment-laden ecosystems on
and beneath the ice sheet (basal ice and incubated basal ice, glacier surface ice, snow, moulin
and cryoconite waters) and subglacial incubation experiments in order to infer the relative
importance of different sources of nitrogen species in runoff. Total Dissolved Nitrogen (TDN),
DIN (nitrate and ammonium) and dissolved organic nitrogen (DON) were quantified in all
samples and exchangeable ammonium associated with suspended sediments (SS-NH$_4^+$) was
analysed in runoff. These data were used to calculate seasonal nitrogen fluxes and yields from
the catchment, and subsequently similar estimates for the GrIS.

**2. Materials and Methods**
**2.1 Field site**
Leverett Glacier is located on the south-west of the GrIS, approximately 300 km north of Nuuk
(Figure 1.; 67.06°N, 50.10°W). The glacier overlies predominantly Precambrian gneiss/granitic
bedrock, typical of much of Greenland as indicated by geological surveys of the non-glaciated
areas bordering the ice sheet (Kalsbeek, 1982;Kalsbeek and Taylor, 1984). The subglacial
sediments are primarily Quaternary deposits (e.g. paleosols) containing fresh organic matter that
were buried during glacial advance in the last few thousands of years following the Holocene
Thermal Maximum when the GrIS margin was positioned tens of kilometres further inland
(Simpson et al., 2009). LG supplies runoff to the Watson River during the summer months, the
largest of three glacially-fed rivers which supply Søndre Strømfjord (Figure 1). Søndre
Strømfjord is the largest fjord system in western Greenland and comprises an inner fjord (up to
275 m deep, 4 km wide and 80 km long) and a shallow outer fjord (<100 m deep, 1km wide and
100 km long) (Nielsen et al., 2010). The inner fjord physical oceanography is influenced by
meltwater, as indicated by a 50-75 m freshwater surface layer (Nielsen et al., 2010).

## 2.2 Sample collection, processing and storage

Two main sampling sites were established in summer 2012: one at the ice sheet margin (11[th]
May – 15[th] July) 1km downstream of the glacier terminus (glacial runoff sampling site, Figure 1,
black dot) and one on the ice sheet surface (8[th] May – 8[th] August) at a moulin located
approximately 35 km from the ice margin (surface meltwater sampling site, Figure 1, red dot).

### 2.2.1   Ice sheet surface sampling

A field camp was established in 2012 in the mid ablation zone at LG at 1030 m elevation, 35 km
from the western margin (66.97°N, 049.27°W). Here, samples of meltwater descending to the ice
sheet bed (Chandler et al., 2013) were collected from the streams feeding a large moulin between
5[th] May and 9[th] August (Day 129 and 222). The discharge of meltwater into this moulin was also
measured (Supplementary Information 1). A range of contextual samples were collected,
including ice containing dispersed cryoconite debris (referred to here as "summer ice") and
cryoconite hole waters. Cryoconite holes are water-filled cylindrical melt holes, formed by
radiation heating of surface sediment and subsequent melting (Podgorny and Grenfell, 1996).
The debris in the base of these holes is termed "cryoconite" which may become distributed over

the glacier surface during melt out of cryoconite holes in summer. Ice samples were melted in clean/sterile Whirl-pak[TM] bags (Nasco) overnight in a warm water bath immediately after collection (melting typically took 2-3 hrs). All meltwater samples were filtered through 47 mm, 0.45 μm cellulose nitrate filters (Whatman[TM]) in a plastic filter unit (Nalgene[TM] PES), pre-rinsed 3 times with sample, and stored in high density polyethylene plastic bottles (Nalgene[TM; ] 30 mL). Samples were frozen immediately after filtration and only thawed out immediately prior to analysis in Bristol. Procedural blanks were processed (n=5) during the course of the sampling season, where deionised water (stored in clean plastic bottles) was treated as a sample.

### 2.2.2   Bulk meltwater sampling

The river draining from the subglacial portal at LG was continuously monitored during the 2012 melt season (May – October) using stage measurements in a stable bedrock section ~2.2 km downstream of the terminus (Hawkings et al., 2014;Hawkings et al., 2015;Cowton et al., 2012;Tedstone et al., 2013;Hawkings et al., 2016). Stage was logged every 5-10 minutes, and converted to discharge using rhodamine dye-dilution experiments (>30 dye tracing experiments were carried out over the season using standard methods (Cowton et al., 2013)). The error in measured discharge determinations is ±10% (Tedstone et al., 2013). Suspended sediment concentrations were calculated as in previous work from bulk meltwater turbidity measurements (Cowton et al., 2012). A turbidity sensor was employed throughout the monitoring period in a similar location to stage measurements. The sensor was calibrated using manual sediment weight samples. Briefly, a recorded amount of meltwater (usually 300 mL) was filtered through a 0.45 μm cellulose nitrate filter (Whatman[TM]), oven dried overnight at 40 °C and weighed.

Bulk meltwater samples were taken approximately 1 km downstream from the LG subglacial portal, at least once a day during the main melt period (May-July). Samples were collected daily at ~10:00 hr, with occasional additional afternoon samples taken at ~18:00 hr, mostly during subglacial outburst events. A 2 L meltwater grab sample was taken in a HDPE Nalgene[TM] bottle (Thermo Scientific[TM]), which had been pre-rinsed 3 times in the meltwater stream. Samples were filtered soon after collection using a Nalgene[TM] reusable PES filtration stack, and a 47 mm 0.45 μm cellulose nitrate filter membrane (Whatman[TM]). Filtered samples were stored in 28 mL HDPE bottles. Procedural blanks were processed (n=10) during the course of the sampling

season, where deionised water (stored in clean Nalgene<sup>TM</sup> HDPE plastic bottles) was treated as a
sample and filtered and bottled accordingly All samples were immediately frozen and stored in
the dark until analysis in Bristol.

### 2.2.3 Basal ice sampling and incubation experiments

Basal ice from the Leverett/Russell Glacier catchment was collected by chain saw from an easily
accessible outcrop of debris-rich basal ice at the ice margin (within 5 km of the main LG bulk
meltwater sampling site, Figure 1) by chain saw (30 x 30 x 30 cm blocks) in spring 2008 and
summer 2010. The outermost ~0.5 m of ice was first removed before the blocks were cut. The
blocks were wrapped in large sheets of pre-combusted foil and stored at $\leq$ -20°C prior to
processing. Sub-samples of the ice were prepared for nitrogen analysis by chipping ~15 x 15 x 5
cm chunks from the main block using a flame sterilised chisel. The outer ~10-30 mm was
removed by rinsing with ultrapure ($\geq$18.2 M$\Omega$cm$^{-1}$) deionized water, and the remaining ice was
transferred into a pre-combusted glass beaker covered with foil. The ice was allowed to melt
inside a laminar flow cabinet (Telstar Mini-H) at room temperature. Icemelt was filtered through
Whatman polypropylene Puradisc™ 0.45 μm syringe filters. Samples for nitrogen species
determinations were stored in clean, thrice-rinsed Nalgene<sup>TM</sup> HDPE bottles. All sediment and
filtered samples were stored in the dark at $\leq$ -20ºC until analytical processing.

Long-term incubation experiments (> 1 yr) were conducted using sediment and meltwater
derived from melted basal ice samples, in order to investigate microbially derived sources of
dissolved nitrogen in a simulated subglacial environment. Three types of experiments were
conducted: 1) Live control experiment with no sediment added, where the solution was
meltwater from basal ice; 2) Live anaerobic experiments (sediment+meltwater from basal ice);
and 3) Live aerobic experiments (sediment+meltwater from basal ice). The control sediment-free
experimental nitrogen concentrations were subtracted from the live (sediment+water)
experiments in order to correct for nitrogen species added from the sampling vessel and the
original basal ice meltwater matrix. Hence, nitrogen concentrations reported are those that have
evolved during the experiment via rock:water contact and *in situ* microbial activity.
Experiments were performed in the dark at 0.1 °C in modified gas-tight 500 mL
borosilicate glass bottles. A sampling port towards the base of the vessel immediately above the
sediment surface was used for water extraction. All incubations contained 100 mL of wet-weight
sediment, 200 mL melt water and 200 mL gas headspace. Control incubation experiments
contained 200 mL ice melt only. Sediment and ice melt (flushed with $O_2$-free-$N_2$ gas) required
for the anaerobic incubations were melted inside a glove-bag filled with $O_2$-free-$N_2$ gas (BOC
Ltd, UK). Meltwater/sediments were later flushed with $O_2$-free-$N_2$ gas for >20 minutes to ensure
that the sediment and water were equilibrated with an oxygen-free atmosphere. The incubations
were sampled ~2 hrs after set-up (T=0 d), on day 4, 109, 190, 294, 382 and 533 and 758 (aerobic
only). At each sampling point, 30 mL (15% of the initial volume) of melt water was removed,
filtered through Whatman polypropylene Puradisc™ 0.45 µm syringe filters and stored at ≤ -
20ºC until analysis. Sampling of the anaerobic incubation experiments were conducted inside a
glove-bag filled with $O_2$-free-$N_2$ gas. All meltwater samples were frozen immediately after
collection and stored frozen prior to analysis for dissolved nitrogen species.

## 2.3 Analytical methods

All meltwater samples were analysed for concentrations of total dissolved nitrogen (TDN),
dissolved inorganic nitrogen (DIN, comprising nitrate and ammonium), with dissolved organic
nitrogen (DON) determined by difference between TDN and DIN. Concentrations of nitrite were
generally below the limit of detection and are not reported. We also analysed ammonium
concentrations associated with suspended sediments in runoff (SS-$NH_4^+$), where this component
is assumed to be bioavailable. The nitrogen content of snow is taken from previous work
conducted in the same catchment (Telling et al., 2012) and from Greenland ice cores (Wolff,
2013). Pre-melt surface glacier ice nitrogen concentrations were taken from previous work
conducted in Leverett glacier catchment (Telling et al., 2012). The detailed sampling and
analytical procedures are provided in the following sections.

### 2.3.1  Nitrate

Nitrate was determined using a Thermo Scientific™ Dionex™ ICS-5000 ion chromatograph
fitted with an IonPac™ AS11-HC-4µm anion-exchange column. A 30 mM KOH eluent
concentration was used, with an injection volume of 0.4 µL and cell temperature of 35°C. The
detection limit of the instrument was 0.08 µM N. The precision of analyses, determined via
analysis of eleven replicate standards at the lower end of the sample range (1.6 µM), was 8.1%.
The accuracy of the machine was determined as -6.4%, using gravimetrically weighed standards
from a 1000 mg $L^{-1}$ certified stock standard (Sigma TraceCERT®). All field nitrate data were
blank corrected using field procedural blanks. The nitrate concentrations within these blanks
were <0.45 µM for surface samples and below the detection limit for runoff samples.

### 211 2.3.2 Ammonium

Ammonium was determined manually using the salicylate spectrophotometric method (Bower
and Holm-Hansen, 1980;Le and Boyd, 2012), adapted for a smaller sample size (1 mL). The
detection limit of the method was 0.6 µM N. The precision of analyses was 4.9 %, calculated
from five replicate standards (1.8 µM). Accuracy was calculated to be +0.3% (from a
gravimetrically diluted certified reference standard, Sigma-Aldrich TraceCERT(R) 1000 mg $L^{-1}$).
Ammonium concentrations in field samples were blank corrected using field procedural blanks,
and were all above the limit of detection. Mean blank correction factors for ammonium were
0.75 µM N for surface samples. Runoff blank corrections were below the detection limit of the
instrument.

### 221 2.3.3 Total Dissolved Nitrogen (TDN)

Total Nitrogen was determined on most runoff samples using a Lachat QuikChem® 8500 Flow
Injection Analyser system, with digestion unit (method number 10-107-04-3-E). The detection
limit of the instrument was 1.4 µM TDN, the precision of analyses was calculated as 11.3% from
six 3.6 µM replicate reference standards (gravimetrically diluted from a certified reference
standard, Sigma-Aldrich TraceCERT(R) 1000 mg $L^{-1}$). Accuracy was determined using the same
reference standards as -0.4%. All TDN data were field blank corrected (2 µM for surface
samples, and no correction for runoff samples since these were below the detection limit of the
instrument).

### 231 2.3.4 Exchangeable $NH_4^+$ in suspended sediment (SS-$NH_4^+$).

232 Measurements were conducted using the method described by (Maynard et al., 2007). Filters

233 containing suspended solids were placed into polypropylene centrifuge tubes and the $NH_4^+$ was

234 then extracted with 10 ml of 2M KCl for 30 minutes on an automatic shaking table (160 rpm).

235 Extracts were decanted into additional centrifuge tubes, centrifuged at 4500 rpm for 5 min, and

236 filtered through 0.45 μm inline Whatman® polypropylene Puradisc filters. When immediate

237 analysis was not possible, they were immediately, frozen (-20°C) until analysis. A second

238 sequential extraction was then performed to extract any residual sediment bound $NH_4^+$. Extracts

239 were analyzed on a Bran and Luebbe Autoanalyzer 3, with a detection limit in extracts of 0.9 μM

240 N, equivalent to 0.09 μM N for a typical sediment mass of 0.1 g. The $NH_4^+$ concentrations from

241 the first and second extracts were combined to give a total $NH_4^+$ for the suspended sediment

242 samples. Dry weights for sediment samples were obtained by washing residual sediment from

243 filters into centrifuge tubes with MQ water, centrifuging at 4500 rpm for 5 min, then repeating

244 with a further MQ wash and centrifuging stage to remove any residual KCl. Sediments were then

245 oven dried (overnight at 40°C) and weighed. This gave concentrations of exchangeable $NH_4^+$ of

246 μg N $g^{-1}$, which were converted into units of μM N $g^{-1}$ and then to μM N by multiplying by the

247 instantaneous suspended sediment concentration (in g $L^{-1}$) at the time of sample collection. SS-

248 $NH_4^+$ fluxes (μM N $s^{-1}$) were subsequently calculated from the product of the $NH_4^+$ concentration

249 and bulk discharge (in L) at the time of sample collection.

250

### 2.4 Flux calculations

### 2.4.1  Nitrogen fluxes from Leverett Glacier

253 Nitrogen fluxes over the entire melt season (May - September) are calculated for LG for the 2012

254 melt season, which was a record melt year in Greenland (Tedesco et al., 2013). Discharge

255 weighted mean concentrations of dissolved nitrogen species and SS-$NH_4^+$ for LG runoff were

256 calculated for the 2012 melt season. Use of discharge weighted mean (DWM) concentrations

257 lowers the mean nitrogen concentrations in bulk meltwaters, since high discharge values are

258 generally accompanied by low nitrogen concentrations. Hence, this method provides a more

259 conservative estimate of nitrogen fluxes. We use minimum and maximum concentrations of

260 nitrogen species to illustrate the potential maximum range of nitrogen fluxes under different

261 hydro-climatological regimes. The product of the DWM, minimum and maximum concentration

of each nitrogen species and the runoff flux for the summer discharge monitoring period in 2012
from LG (2.2 km$^3$, Supplementary Figure 5) generated the total seasonal fluxes of these nitrogen
species. We did not measure the particulate organic nitrogen (PON) concentrations in runoff, and
in previous years these concentrations have been below the detection limit of standard analytical
methods. However, we did calculate the SS-NH$_4^+$ fluxes in the same manner as the dissolved
nitrogen species. Total fluxes of dissolved and SS-NH$_4^+$ in LG runoff during the 2012 melt
season are presented in Table 2. Errors on these estimates due to discharge uncertainty and
catchment area are of the order of ±10% and ±25% respectively (Tedstone et al., 2013;Cowton et
al., 2012), giving a combined uncertainty of ±27%.

### 2.4.2  Nitrogen fluxes from the Greenland Ice Sheet

Currently, there are no other seasonal time series of nitrogen concentrations in runoff from large
Greenland outlet glaciers. Hence, nitrogen concentrations in LG runoff are used in order to
generate order of magnitude flux estimates for nitrogen associated with Greenland freshwater
export. We base our calculations upon the premise that LG is representative of large areas of the
GrIS, for several reasons. First, LG displays a high altitudinal range (250 – 1510 m a.s.l.) and
extends for > 80 km inland, like many large Greenland outlets. Hence, nitrogen supply from
snow and ice melt are likely to be representative of other large catchments draining the ice sheet.
Second, microbial processes (e.g. nitrogen fixation, nitrification, organic matter mineralisation),
which are thought to generate approximately half of the ice sheet nitrogen in runoff (via DON,
nitrate and ammonium), are reported from a wide range of other glacial systems worldwide
including the Greenland Ice Sheet (Hodson et al., 2005;Boyd et al., 2011;Telling et al., 2012), a
reflection of the ubiquitous nature of microbial ecosystems upon glacier surfaces and at glacier
beds. Third, the bedrock geology at LG is representative of large areas of the GrIS (see Section
2.1). This suggests that the drivers for nitrogen export at Leverett Glacier are likely to be
applicable to other large catchments, which account for the bulk of the freshwater flux from the
ice sheet to the oceans. Our approach is widely employed for calculating solute fluxes from ice
sheet systems where datasets are sparse due to the difficulty of making measurements (Wadham
et al., 2010;Bhatia et al., 2013;Lawson et al., 2013;Hawkings et al., 2014;Hawkings et al., 2016).
Fluxes of nitrogen from the GrIS are calculated from the product of DWM, minimum and
maximum concentrations of the different nitrogen species at LG glacier (Table 3) and the total
ice sheet runoff flux for 2012 and the mean runoff flux of 2000-2011 (Tedesco et al., 2013)
(Table 3). The latter is modelled using the MAR regional climate model. ==Errors for meltwater==
==runoff determinations via the MAR model are estimated 10% (Vernon et al., 2013). Hence, we==
==might expect similar uncertainty to propagate to nutrient flux determinations.== We also estimate
the potential nitrogen fluxes exported to the ocean by iceberg calving, which have a potential far-
field influence within the open ocean (Syvitski et al., 2001;Smith Jr. et al., 2013). Iceberg
nitrogen fluxes are taken to be the product of the iceberg freshwater flux and mean nitrogen
concentrations in Greenland ice cores (Table 3). We employ a freshwater flux for Greenland
icebergs of 600 km$^3$ a$^{-1}$, based upon approximate average values for the last decade (Bamber et
al., 2012). We assume that the mean concentrations of nitrogen in icebergs are similar to those
reported in Greenland ice cores (Wolff, 2013), which are also in line with those reported in LG
catchment (Telling et al., 2012). This is a conservative estimate, since additional nitrogen supply
is likely associated with sediments entombed within icebergs. Results from this work indicate
that the SS-NH$_4$ content of ice containing even trace amounts of debris may display elevated
nitrogen concentrations which are five times higher than in ice with no debris (Table 1).


## 3.  Results and Discussion

### 3.1 Sources of nitrogen in runoff

The LG runoff time series demonstrates that the GrIS provides a continuous supply of nitrogen
to downstream ecosystems throughout the main melt period (Figure 2). Concentrations of TDN
are significant (1-10μM) and mean nitrate concentrations (1.8 μM) alone are higher than those
reported in surface ocean and fjord waters (<0.1-1μM) in western Greenland in summer (Nielsen
and Hansen, 1999;Arendt et al., 2010;Hopwood et al., 2016). Higher concentrations of nitrate are
observed in deeper ocean waters, but upward diffusion and advection are often limited by a
stratified water column during the summer months (Arendt et al., 2010). DIN, which is readily
available to marine phytoplankton, accounts for half of the TDN in LG runoff, supplemented by
SS-NH$_4^+$ from the ice sheet bed. A component (~50%) of the DIN measured in LG runoff
originates from natural and anthropogenic atmospheric sources, via melting of snow and ice

(Wolff, 2013) (Table 1). LG drains a large catchment (active hydrological catchment area = 600 km$^2$ ±25% (Cowton et al., 2012)) with a high altitudinal range (extending to >1500 m a.s.l.). New moulins open up and surface lakes drain with snow line retreat (Bartholomew et al., 2011), providing a mechanism by which new sources of DIN are fed to runoff. Water fluxes control the overall nitrogen flux, which rises through summer to attain high values during the sampling period in mid-July (Figure 2). The bulk runoff chemical sampling record did not extend beyond this point. However, we assert that runoff nitrogen fluxes will continue to be high in late July/early August, as evidenced by the sustained high fluxes of nitrogen species in moulin waters up until 9[th] August (Day 222, Suppl. Figure 5). This is significant given the reported nitrogen limitation of fjord and marine phytoplankton in mid-summer, once the water column becomes more stratified and deep marine sources of nitrogen become more inaccessible (Rysgaard et al., 1999;Budeus and Schneider, 1995).

A striking feature of the runoff dataset is the factor of four increase in concentrations of TDN in LG runoff (4.5 µM, 5.7 µM including SS-NH$_4^+$) compared with those in snow and ice (<1 µM), reflecting enhancements in dissolved organic nitrogen, ammonium and nitrate (Figure 3, Table 1). Similar findings have been reported at small valley glaciers (Hodson et al., 2008), and imply the acquisition of significant quantities of nitrogen within the glacier. A substantial proportion of this enhancement must occur in sedimentary environments at the ice sheet bed, as indicated by a significant association between TDN in moulin waters and bulk runoff, but a positive intercept of 2.5 µM (Figure 4). A range of possible sources exist for this additional nitrogen in runoff. Our wider contextual survey of the nitrogen content of basal and surface ice and meltwaters and subglacial incubation experiments allows us to conjecture on these sources. For nitrate, enhancement is likely to occur in the subglacial environment, since nitrate concentrations in moulin waters and snow/ice are similar (Table 1). The basal regions of ice sheets are viable habitats for microbial life and previous work has demonstrated the activity of nitrifying bacteria, which transform ammonium to nitrate, at small Alpine valley glaciers (Boyd et al., 2011;Wynn et al., 2007) and Subglacial Lake Whillans in Antarctica (Christner et al., 2014). In support of this, long-term incubation experiments using LG subglacial sediments (Figure 5) show the release of up to 5 µM nitrate under aerobic conditions in live sediments and an absence of this production in live controls (no sediment) and under anaerobic conditions. The

simultaneous removal of ammonium ions is consistent with nitrification as the source of this nitrate, likely in more aerobic subglacial channel-marginal sedimentary environments.

The enhancement of DON concentrations in moulin waters relative to snow and ice, and in runoff relative to moulin waters is also significant (independent t-test, p=0.05) and suggests the acquisition of DON in surface and basal ecosystems respectively. This is consistent with previous work that has suggested the presence of a significant nitrogen-rich component to dissolved organic matter exported from glacier ecosystems in runoff (Hood et al., 2009;Lawson et al., 2014;Bhatia et al., 2013). Likely surface sources are cryoconite holes and debris-rich ice, which display elevated DON concentrations relative to pre-melt ice and snow (Figure 2, Table 1). These debris-laden environments support diverse microbial communities, which actively fix carbon dioxide from the atmosphere (Stibal et al., 2012). We assert that mineralization of organic matter in such environments generates the elevated DON concentrations in surface waters by microbial activity or by leaching from allochthonous organic matter in debris. The factor of two enhancement in DON concentrations in runoff relative to moulin waters reflects an even greater subglacial input of these nitrogen species. It is notable also that, while ammonium concentrations in bulk runoff are generally low (< detection limit at 0.6 μM), both DON and ammonium concentrations in runoff are often elevated during subglacial outburst events, rising to up to 3 and 6 μM respectively (Figures 2 and 6). These events are known to expel long-term stored meltwaters and sediments from beneath the ice sheet in response to surface lakes drainage (Bartholomew et al., 2011),. A subglacial source of these waters is clearly evident from elevated sulphate concentrations, which rise during outburst events. Sulphate ions are uniquely generated in inefficient distributed drainage pathways at the glacier bed, where comminution of the underlying bedrock releases highly reactive iron sulphide minerals to meltwaters. These oxidise rapidly to give sulphate (Tranter et al., 1993). The elevated runoff DON and ammonium during such events implies a source in subglacial sedimentary ecosystems. Long-term incubation experiments presented in Figure 5 strongly support a subglacial source for DON but do not show elevated concentrations of ammonium in live experiments. We propose that the subglacial acquisition of DON reflects *in situ* microbial activity, as reported beneath smaller valley glaciers (Hodson et al., 2005). The low dissolved organic carbon (DOC):DON ratio in runoff (mean=9.5, DOC data from (Hawkings, 2015)) is similar to other world glaciers (Hood and Scott, 2008) and is consistent with a microbial source for DON. It contrasts with the higher mean DOC:DON

ratios for Arctic rivers (mean=48) which include a greater terrestrial contribution (Lobbes et al.,
2000). These findings support the notion that dissolved organic matter exported from the GrIS
may be highly bioavailable to marine bacteria (Lawson et al., 2014;Lawson et al., 2013;Bhatia et
al., 2010), as has been suggested for glacier systems elsewhere (Hood et al., 2009).
The subglacial source of the enhanced ammonium concentrations in long-term stored
subglacial waters released during outburst events is less clear. Ammonium concentrations in
basal ice and outburst waters were relatively high (mean = 2-3 μM) (Table 1; Figure 6) but our
subglacial incubation experiments time series showed no significant enhancement of ammonium
from initial concentrations over time (Figure 5). Enhancement of ammonium concentrations in
long-term stored subglacial meltwaters have been documented previously in Antarctic Subglacial
Lake Whillans, inferred to reflect microbial mineralisation (Christner et al., 2014). There are
several potential reasons for the static ammonium concentrations during our laboratory
experiments. It may reflect the difficulty of replicating microbial processes under laboratory
conditions, together with elevated starting ammonium concentrations at t=0 in experiments
(basal ice mean $NH_4^+$=2.7 μM). Second, it may indicate that the subglacial process that generates
ammonium ions is not modelled well by laboratory experiments. For example, ammonium may
be released to solution directly or indirectly by crushing of the underlying bedrock (Dixon et al.,
2012), as occurs for other species such as hydrogen (Telling et al., 2015). Greenland gneiss
contains very small concentrations of nitrogen (9 μg N g[-1] (Holloway and Dahlgren, 2002)), but
glacial crushing and release of this nitrogen from bedrock as ammonium has the potential to
generate concentrations an order of magnitude higher concentrations of ammonium that those
observed in incubation experiments. Overall, data presented here suggests the operation of a
suite of diverse mechanisms that supply nitrogen species from snow and icemelt, enhancing them
in supra- and subglacial ecosystems prior to meltwaters being evacuated at the ice margin. This
is consistent with recent work in Alpine regions that clearly demonstrated the potential for
glacier-fed catchments to display enhanced nitrogen concentrations in runoff relative to
snowmelt-fed systems (Saros et al., 2010).

### 3.2 Fluxes of nitrogen from Leverett Glacier

Total dissolved nitrogen (including SS-$NH_4^+$) fluxes from LG in summer are on average 0.14 t a[-]
[1] (Table 2). The estimated TDN yields for the Leverett Glacier catchment arising from this flux
are 236 kg km$^{-2}$ (164 kg m$^{-2}$ excluding SS-NH$_4^+$), which is an order of magnitude higher than the
typical annual TDN yields measured in large Arctic rivers (36-81 kg km$^{-2}$) (Holmes et al., 2012).
This high yield largely arises from the high specific water yield at LG (3.7 x 10$^6$ m$^3$ km$^{-2}$ a$^{-1}$), in
comparison to the water yield (July-October) of the largest Arctic rivers, which is two orders of
magnitude lower (9.3 x 10$^4$ m$^3$ km$^{-2}$ a$^{-1}$, calculated from a water flux for the six largest Arctic
rivers of 1011 km$^3$ a$^{-1}$ from July to October and a gauged catchment area of 10.9 x 10$^6$ km$^2$)
(Holmes et al., 2012). This implies that there is a much higher continuous flux of dissolved
nitrogen species per unit area from the ice sheet in summer than from high Arctic River
catchments, reflecting the acquisition of dissolved N species from both melting snow and ice on
the surface and sedimentary environments at the ice sheet bed.

## 3.3 Fluxes of nitrogen from the Greenland Ice Sheet

The estimated summer mid-range TN flux (including SS-NH$_4^+$) from the GrIS is ~27 Gg (2000-
2011) and 43 Gg (2012) (Table 3) using discharge weighted mean concentrations, with potential
flux ranges of 7-86 and 11-137 for 2000-2010 and 2012 respectively. The mean values are of a
similar order of magnitude to a large Arctic river (the average TDN flux for the Lena, Yenisey,
Ob Rivers, July-October is 41 Gg (Holmes et al., 2012)). The glacial nitrogen fluxes largely
supply different ocean basins to the Arctic rivers (Bamber et al., 2012;Holmes et al., 2012). We
contend that ice sheet derived nitrogen fluxes are likely to rise with enhanced melting in a
warmer climate and could, therefore, stimulate increased primary production in downstream
coastal ecosystems. Evidence from a single melt year suggests that within-season fluxes of
nitrogen species rise exponentially with increasing glacial water fluxes (Figure 2). The degree of
future nitrogen flux increase in warm melt years, however, is difficult to predict. The
atmospheric nitrogen flux (largely as DIN) are likely to scale with increasing melt volumes as
has been suggested else (Hawkings et al., 2015). However, the magnitude of increase will
depend upon the availability of glacial ice and snow from post-industrial times, since these
display elevated atmospheric DIN compared with pre-industrial ice (Olivier et al., 2006). DON
and non-atmospheric ammonium fluxes might also be expected to increase as the zone of melting
expands and there is more extensive contact of meltwater with organic matter in surface and
subglacial ecosystems.

441   The impact of present and future nitrogen fluxes upon fjord and coastal marine

442 ecosystems around Greenland is unknown, and requires further study. The input of nutrients

443 associated with Greenland icebergs and runoff may sustain elevated primary productivity beyond

444 the spring phytoplankton bloom, and offers one possible explanation for the reported mid-

445 summer phytoplankton bloom in Western Greenland (Frajka-Williams and Rhines, 2010;Nielsen

446 and Hansen, 1999). Nitrogen limitation is common in fjord and coastal waters in summer, and

447 hence any increase in DIN supply has the potential to enhance primary productivity.

448

## 4 Conclusions

450 In summary, our findings at Leverett Glacier suggest that large glacial outlet glaciers draining

451 the Greenland Ice Sheet provide a continuous source of dissolved nitrogen in runoff through the

452 summer months, a proportion of which is likely to originate from microbial ecosystems on and

453 beneath the ice. The degree to which these nitrogen fluxes are modified by proglacial processes

454 is unknown, as are the potential impacts upon fjord and coastal marine biological productivity.

455 However, phytoplankton in coastal Greenlandic waters often become limited by nitrogen

456 availability by mid-summer, when the glacial nitrogen flux to coastal waters is highest. TDN

457 yields from Leverett Glacier are an order of magnitude higher than those reported for Arctic

458 rivers, a reflection of the high surface melt rates (and hence water fluxes) and continuous

459 nitrogen supply from several sources within the ice sheet. Estimated fluxes of nitrogen from the

460 ice sheet are similar in magnitude to those of a large Arctic river. Our findings suggest that a

461 melting GrIS may be an important source of nitrogen to downstream coastal ecosystems, and that

462 these nitrogen fluxes are likely to increase in a warming climate.

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

## 6  Acknowledgments

This research is part of the UK Natural Environment Research Council, NERC funded DELVE
project (NERC grant NE/I008845/1). It was also funded by NERC grants NE/E004016/1 (to J. L.
Wadham), NE/F0213991 to (P.W. Nienow) and a NERC CASE studentship to E. C. Lawson
(NERC DTG/GEOG   SN1316.6525) co-sponsored by Dionex Corporation (part of Thermo
Fisher Scientific) and a NERC PhD studentship to J. Hawkings. A. Tedstone was funded by a
NERC studentship and MOSS scholarship. P.W. Nienow was supported by grants from the
Carnegie Trust for University of Scotland and The University of Edinburgh Development Trust.
Additional support was provided by the Leverhulme Trust, via a Leverhulme research fellowship
to J.L. Wadham. We thank all of those assisted with fieldwork at LG, and to Dr Fanny Monteiro
who provided comments on an earlier draft. The work was also supported by the Cabot Institute
at the University of Bristol.






## 7 Tables

**Table 1** Mean concentrations of nitrogen species reported in LG runoff (including discharge weighted mean, DWM for TDN species), in comparison to those in moulin waters, surface ice (pre-melt and post-melt "Summer ice", where the latter samples were at the melting temperature and contained dispersed debris), snow, cryoconite water and basal ice (a-(Telling et al., 2012), b-(Wolff, 2013))

| | $NO_3^-$ (μM) | | | $NH_4^+$ (μM) | | | DIN (μM) | | | DON (μM) | | | TDN (μM) | | |
|---|---|---|---|---|---|---|---|---|---|---|---|---|---|---|---|
| | mean | SD | n | mean | SD | n | mean | SD | n | mean | SD | n | mean | SD | n |
| Bulk runoff-dissolved | 1.8 | 1.2 | 62 | 0.4 | 0.6 | 62 | 2.2 | 1.4 | 62 | 2.3 | 1.5 | 62 | 4.5 | 2.3 | 62 |
| Bulk runoff-dissolved, DWM | 1.1 | - | - | 0.3 | - | - | 1.4 | - | - | 1.7 | - | - | 3.2 | - | - |
| Bulk runoff-sediment bound | n.d. | n.d. | - | 1.2 | 0.6 | 39 | 1.2 | n.d. | - | n.d. | n.d. | - | 1.2 | n.d. | - |
| Bulk runoff-sediment bound DWM | n.d. | n.d. | - | 1.4 | - | 39 | n.d. | n.d. | - | n.d. | n.d. | - | n.d. | n.d. | - |
| Moulins (same time period) | 0.7 | 1.4 | 28 | 0.6 | 0.5 | 28 | 2.0 | 1.2 | 28 | 1.1 | 1.3 | 28 | 2.2 | 1.4 | 28 |
| **SURFACE** | | | | | | | | | | | | | | | |
| Pre-melt ice[a] | 0.59 | 0.14 | 6 | 0.3 | 0.1 | 6 | 0.9 | 0.3 | 6 | 0.0 | 0.0 | 6 | 0.6 | 0.1 | 6 |
| Snow[a] | 1.03 | 0.17 | 3 | 0.45 | 0.0 | 3 | 1.1 | 0.2 | 3 | 0.0 | 0.0 | 3 | 1.02 | 0.14 | 3 |
| GrIS ice cores[b] | 0.97 | n.d. | - | 0.45 | n.d. | - | 1.4 | n.d. | - | n.d. | n.d. | - | n.d. | n.d. | - |
| Summer ice | 0.64 | 0.42 | 7 | 0.6 | 0.6 | 7 | 1.3 | 0.9 | 7 | 3.0 | 2.6 | 7 | 2.9 | 2.1 | 7 |
| Cryoconite meltwater | 1.4 | 0.4 | 6 | 1.1 | 1.3 | 6 | 1.7 | 0.9 | 6 | 0.7 | 0.4 | 6 | 2.4 | 1.1 | 6 |
| **SUBGLACIAL** | | | | | | | | | | | | | | | |
| Basal ice | 1.5 | 0.0 | 6 | 2.7 | 0.1 | 6 | 3.7 | 0.1 | 6 | 12 | 1.3 | 6 | 15 | 1.3 | 6 |
| Incubations (aerobic) | 1.4 | 2.1 | 7 | 2.5 | 2.2 | 7 | 3.9 | 2.5 | 7 | 3.36 | 2 | 7 | 7.1 | 1.7 | 7 |
| Incubations (anaerobic) | 1.03 | 1.4 | 6 | 0.76 | 0.2 | 6 | 1.8 | 1.3 | 6 | 1.79 | 1 | 6 | 5.3 | 1.4 | 6 |





**Table 2** Estimates of seasonal fluxes of total dissolved (TDN) and particulate nitrogen (SS-
$NH_4^+$) species (total nitrogen=TN) from Leverett Glacier and the Greenland Ice Sheet in 2000-
2010 and 2012 (values marked with an asterisk were below the analytical limit of detection but
and hence, are purely indicative)

| Glacial Runoff: Leverett Glacier (LG) | | | | | | | |
|---|---|---|---|---|---|---|---|
| [b]LG Water Flux ($km^3\ a^{-1}$) (2012) | 2.2 | | | | | | |
| Concentration LG ($\mu M$) | **TDN** | **DIN** | **DON** | **$NO_3^--N$** | **$NH_4^+-N$** | **$SS-NH_4^+-N$** | **TN+SS-NH4+** |
| min | 0.9* | 0.5 | 0.1 | 0.1 | 0.4* | *0.31 | 1.2 |
| DWM | 3.2 | 1.5 | 1.7 | 1.1 | 0.3 | 1.4 | 4.6 |
| max | 11 | 7.5 | 6.3 | 5.1 | 2.4 | 3.7 | 15 |
| Flux LG ($t\ a^{-1}$): 2012 | **TDN** | **DIN** | **DON** | **$NO_3^--N$** | **$NH_4^+-N$** | **$SS-NH_4^+-N$** | **TN+SS-NH4+** |
| min | 28 | 15 | 3 | 3 | 12 | 10 | 37 |
| DWM | 99 | 46 | 52 | 34 | 9 | 43 | 142 |
| max | 339 | 231 | 194 | 157 | 74 | 114 | 453 |
| Yield, using DWM (kg N $km^2\ a^{-1}$) | 164 | 77 | 87 | 56 | 15 | 72 | 236 |

b-measured water flux from Leverett Glacier (this manuscript)
**Table 3** Estimates of seasonal fluxes of total dissolved (TDN) and particulate nitrogen (SS-$NH_4^+$)
species (total nitrogen=TN) from the Greenland Ice Sheet in 2000-2010 and 2012

| Glacial Runoff: Greenland Ice Sheet | | | | | | | |
|---|---|---|---|---|---|---|---|
| [a] GrIS Water Flux ($km^3\ a^{-1}$) (2000-2011) | 418 | | | | | | |
| [a] GrIS Water Flux ($km^3\ a^{-1}$) (2012) | 665 | | | | | | |
| | **TDN** | **DIN** | **DON** | **$NO_3^--N$** | **$NH_4^+-N$** | **$SS-NH_4^+-N$** | **TN+SS-NH4+** |
| **Flux GrIS ($Gg\ a^{-1}$): 2000-2010** | | | | | | | |
| Min | 5.3 | 3 | 1 | 1 | 2 | 2 | 7.1 |
| DWM | 19 | 9 | 10 | 6 | 2 | 8 | 27 |
| Max | 64 | 44 | 37 | 30 | 14 | 22 | 86 |
| **Flux GrIS ($Gg\ a^{-1}$): 2012** | | | | | | | |
| Min | 8.4 | 4.7 | 0.9 | 0.9 | 3.7 | 2.9 | 11 |
| DWM | 30 | 14 | 16 | 10 | 2.8 | 13 | 43 |
| Max | 102 | 70 | 59 | 47 | 22 | 34 | 137 |
| **Ice Discharge Greenland Ice Sheet** | | | | | | | |
| [a]GrIS Iceberg Discharge ($km^3\ a^{-1}$) | ~600 | | | | | | |
| | **TDN** | **DIN** | **DON** | **$NO_3^--N$** | **$NH_4^+-N$** | **$SS-NH_4^+-N$** | **TN+SS-NH4+** |
| | 1.4 | 1.4 | 0 | 0.97 | 0.45 | n/a | 1.4 |
| | 12 | 12 | 0.0 | 8 | 4 | n/a | 12 |
| **Arctic River Discharge** | | | | | | | |
| [c]Arctic River mean summer water flux ($km^3\ a^{-1}$) | 169 | | | | | | |
| | **TDN** | **DIN** | **DON** | **$NO_3^--N$** | **$NH_4^+-N$** | **$SS-NH_4^+-N$** | **TN+SS-NH4+** |
| [c]Concentration Arctic Rivers ($\mu M$) | 14 | 2.7 | 12 | 2.0 | 0.7 | n/a | n.d. |
| [d]Mean summer flux Arctic Rivers ($Gg\ a^{-1}$) | 41 | 8.8 | 33 | 7.7 | <0.5 | n.d. | n.d. |



## 8   Figure Captions

**Figure 1** Map showing the study area, including the location of Leverett Glacier runoff sampling station (white dot), surface sampling site (red dot) ==and the basal ice sampling location (brown dot), together with Søndre Strømfjord and the Watson River, into which runoff from Leverett Glacier drains.==

**Figure 2 Times series of nitrogen species in LG runoff from the 2012 melt season** depicting concentrations of a) bulk meltwater suspended sediment and sediment-bound ammonium (SS-$NH_4^+$) b) TDN and DON, c) dissolved nitrate and ammonium, and instantaneous fluxes of, d) SS-$NH_4^+$ (bulk meltwater discharge, Q, is also shown), e) TDN and DON and f) dissolved nitrate and ammonium. Vertical dotted lines (left to right) indicate 1[st] May, 1[st] June and 1[st] July, 2012. ==The grey shaded bars reflect inferred subglacial outburst events (Hawkings et al., 2014)==

**Figure 3 Associations between TDN and DIN** in a) runoff, moulin waters, snow and pre-melt ice, where data on snow and pre-melt ice are from (Wolff, 2013;Telling et al., 2012) and b) runoff and glacier surface ecosystems (cryoconite holes, summer ice including dispersed debris) and subglacial ecosystems (basal ice and meltwaters sampled from anaerobic/aerobic long-term subglacial (SG) incubation experiments). A line indicates ratios of 1 for TDN/DIN where the TDN content of samples is entirely comprised of DIN. Samples that plot below this line have a dissolved organic nitrogen component. All samples have been blank corrected and error bars reflect the uncertainty of nutrient analyses given known precision and accuracy.

**Figure 4** Association between the TDN concentrations measured simultaneously at the moulin and runoff monitoring sites (the correlation is significant at the 99% confidence level). Insets show the same data (excluding runoff samples) at low concentrations (< 6 μM).

**Figure 5** Time series of dissolved nitrogen concentrations measured in a) Live aerobic, b) Live anaerobic c) Sediment-free aerobic and d) Sediment-free anaerobic incubation experiments (note the difference in scale for the y axis between a) and b)-d).

**Figure 6** Times series of a) bulk discharge, b) concentrations of sediment-bound $NH_4^+$ (P-$NH_4^+$)
and dissolved sulphate and c) concentrations of DON and dissolved $NH_4^+$ in runoff measured
during the first (and main) subglacial outburst event during the 2012 season.
**Figure 1**

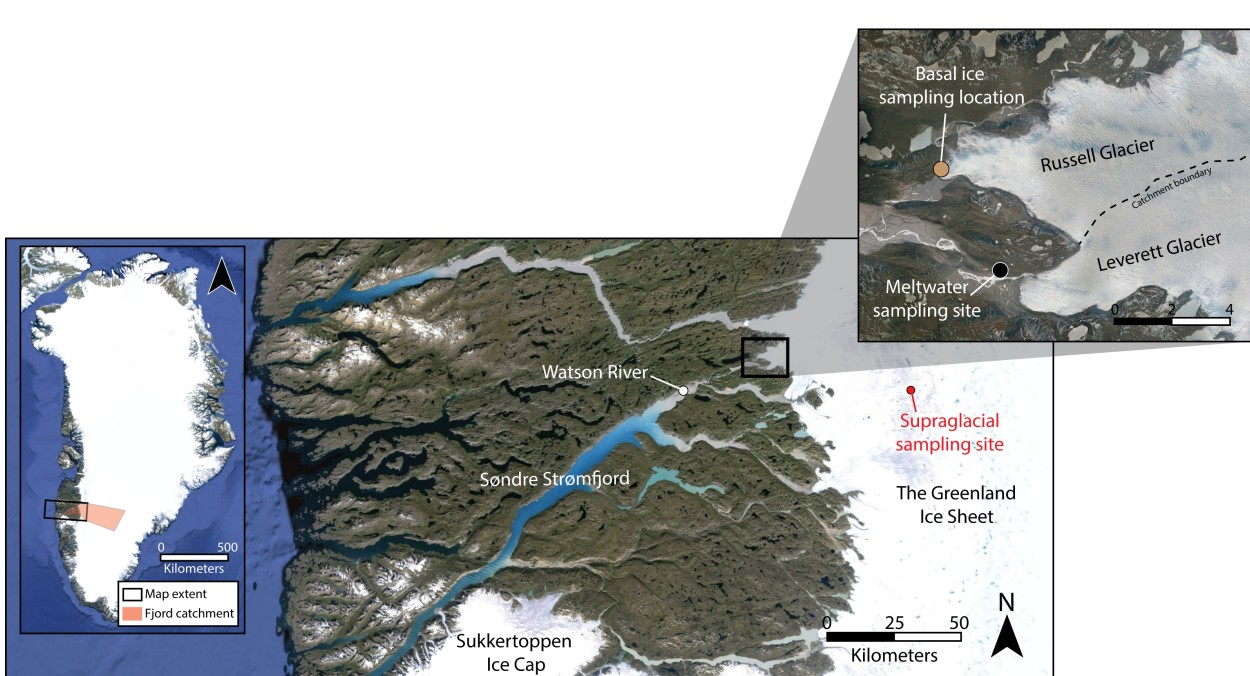
















**Figure 2**

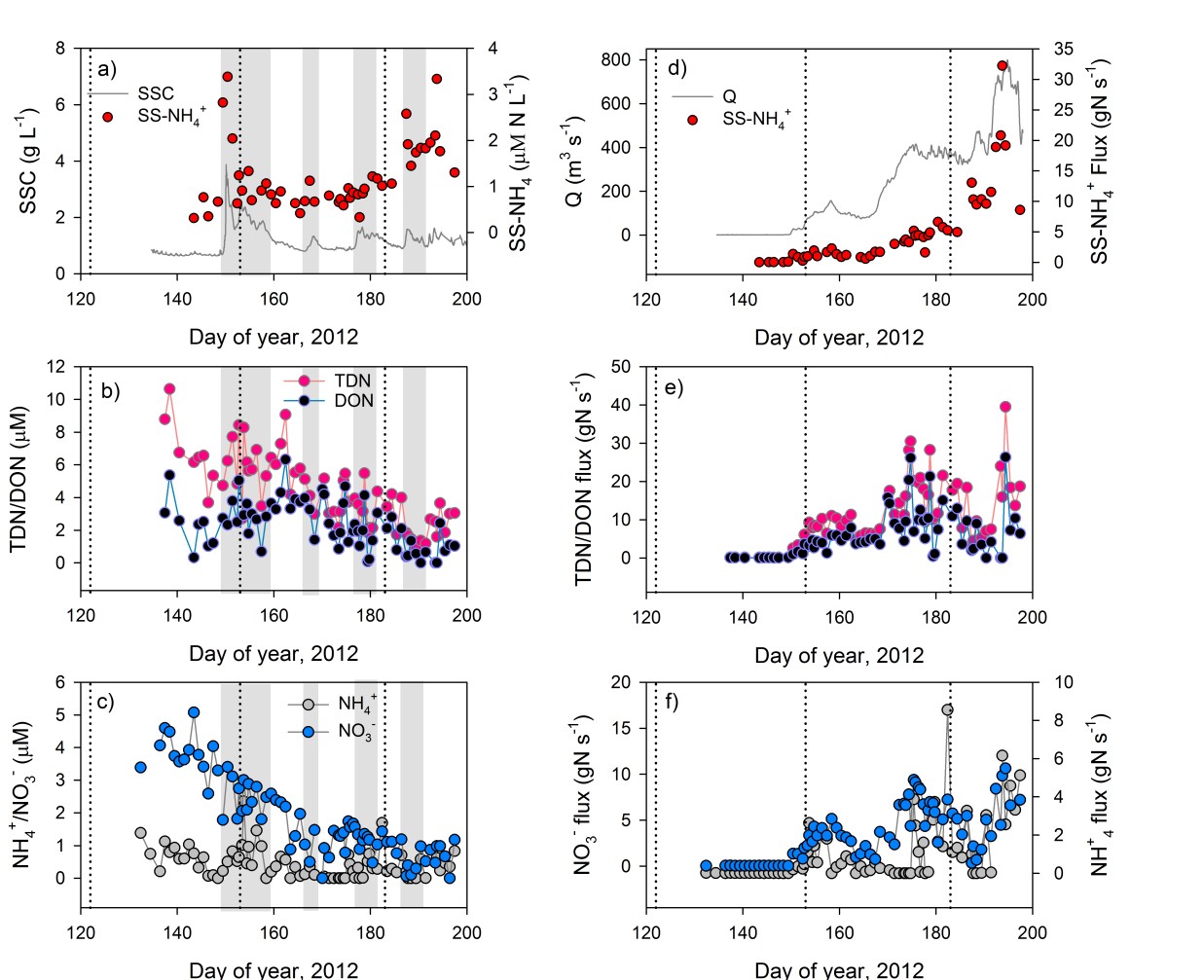



**Figure 3**

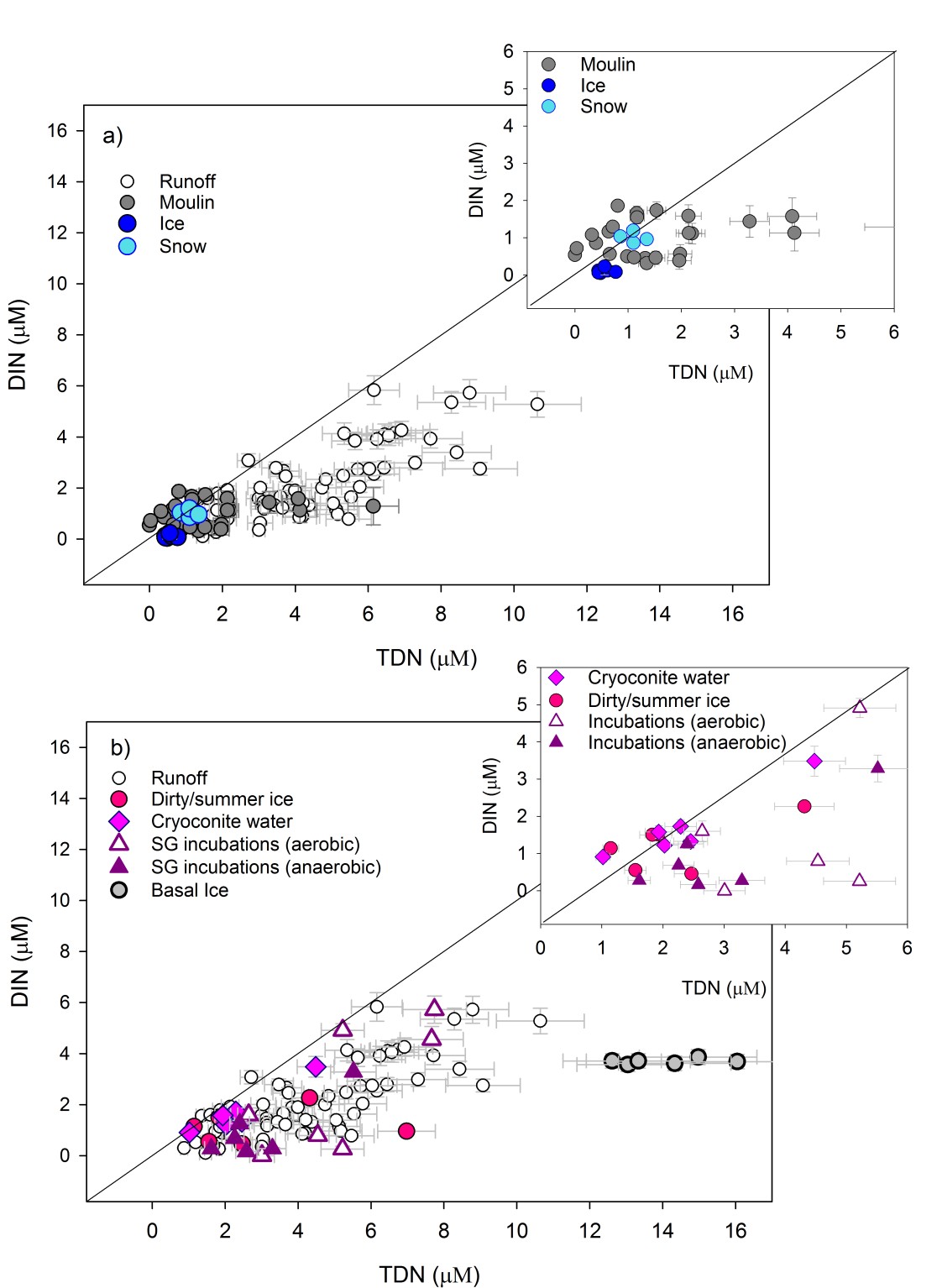



**Figure 4**

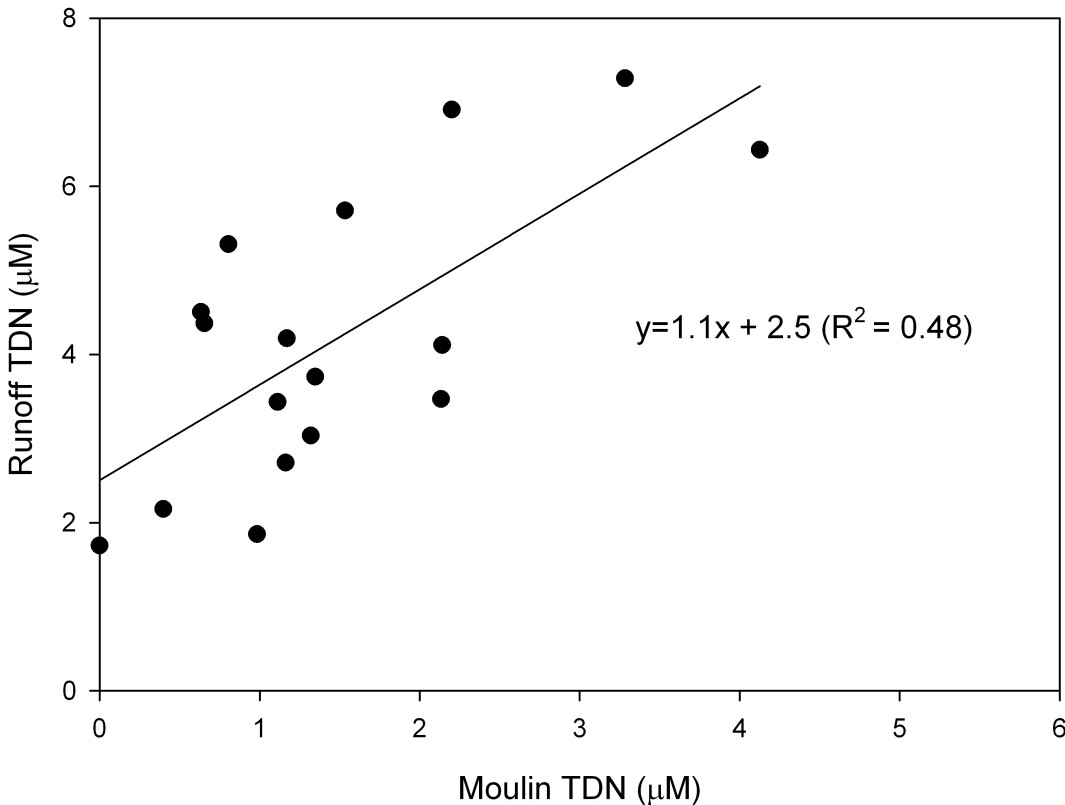















**Figure 5**

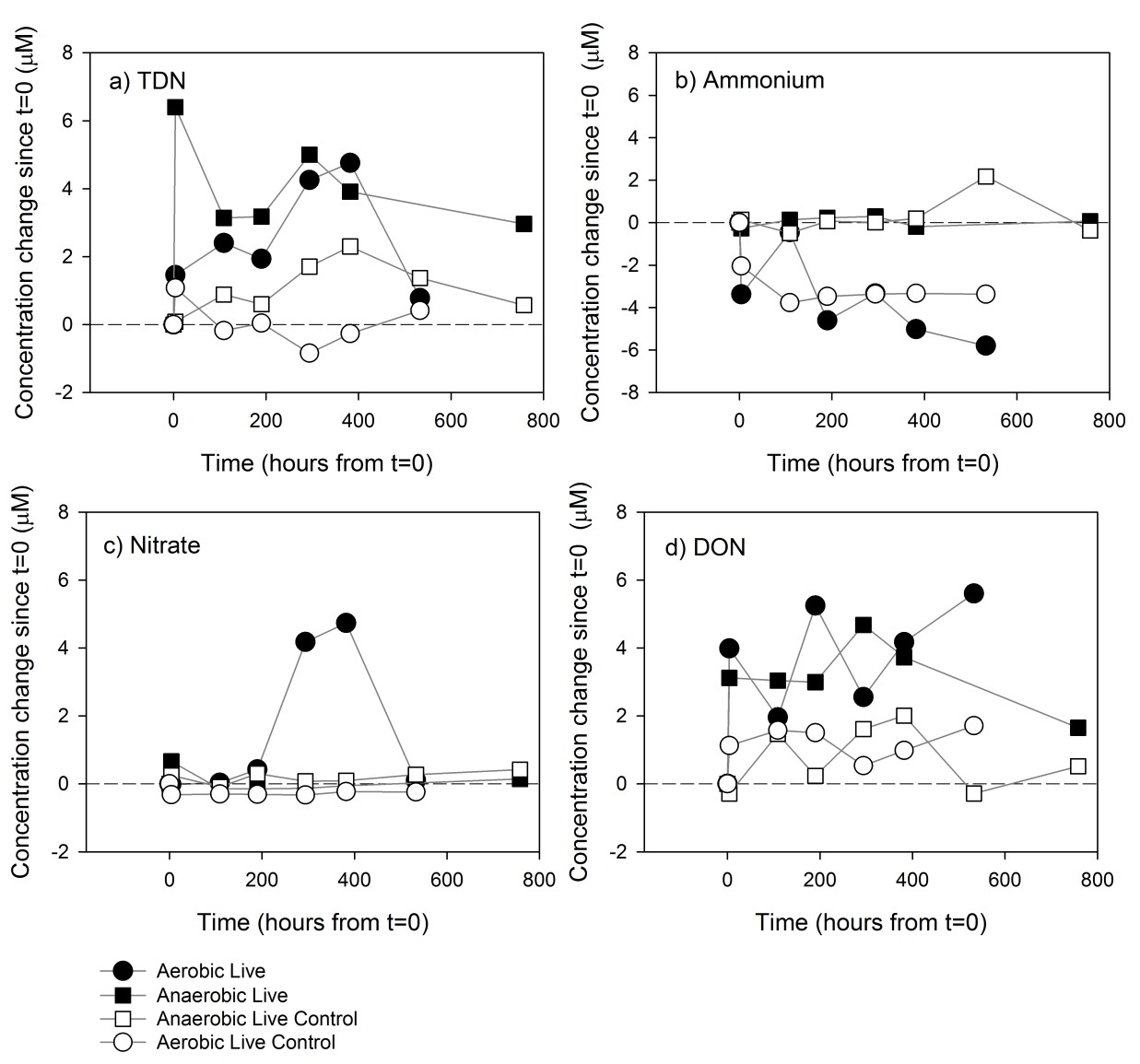


**Figure 6**

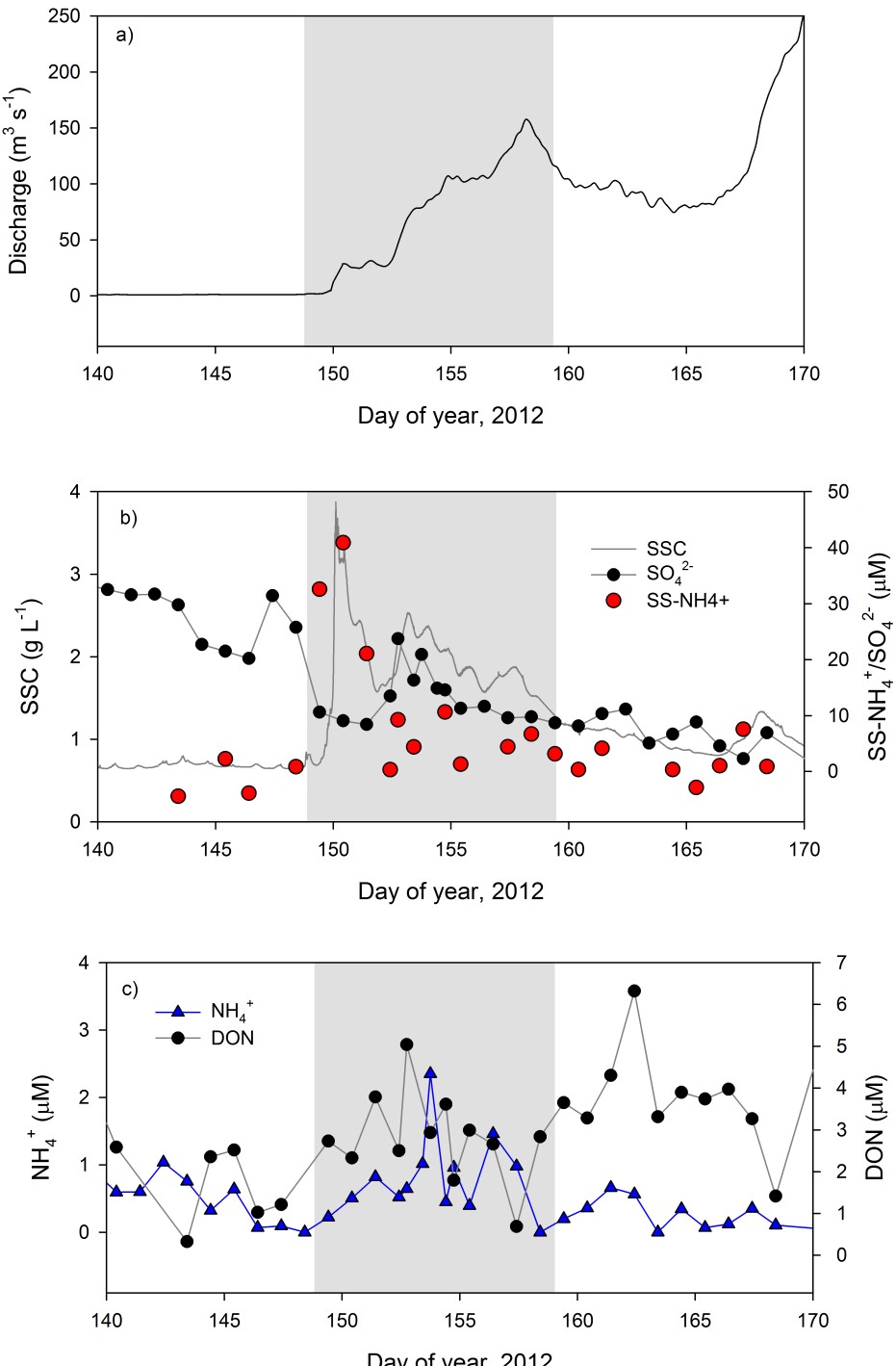


