# Peer review of "Sources, cycling and export of nitrogen on the Greenland Ice"

_Biogeosciences, 2015_

## Short Comment (SC1) · 29 Feb 2016

This review was jointly written by Amy Ritchie, Jenny Davidson, Sally Jenzen, Peter Foster and Cameron Campbell, students at the University of Edinburgh, with some input and supervision from Dr Ed Mitchard.

— The authors present a novel dataset – we believe it is the first time that nitrogen export from a glacier in Greenland has been quantified. We believe this is an important topic, with relevance for the wider environment especially in the context of rapid climate change in this region. The methodology followed appears sound, and the dataset presented is certainly useful and provides a step forward in information on the Greenland Ice Sheet. However, the extrapolation to the whole Ice Sheet may be questionable, and we have some five major specific concerns that we would like the authors to address,

detailed below. Further, we have some specific, more minor comments.

1) Introduction: While the introduction to this paper is adequate, it is very short and we have identified some potential improvements that could be made. We feel the introduction is somewhat brief and lacks explanation for why the study site was chosen nor the background behind the study and the methods used.

Specifically, you state that the paper will examine the sources and cycling of nitrogen on the Greenland Ice Sheet. However, while you state that your study site will be the Leverett Glacier in southwest Greenland, and refer to it throughout, there is no evidence as to why this particular area is representative of the whole of the Ice Sheet. It would be beneficial to include some evidence of the Leverett Glacier's importance in the context of nutrient cycling in Greenland, and why this site was chosen. It would be useful to put a sentence in the Introduction or Study Site Description giving the proportion of the Greenland Ice Sheet that is drained by this specific glacier.

Not only this, but it would be beneficial to provide more context and relevance of the general issues explored in the paper. You state "there is comparatively little data on nitrogen sources and cycling on the Greenland Ice Sheet, which is likely to be important as a nutrient source to downstream fjord and marine ecosystems." While this provides some background to the specifics of the Greenland ecosystem, this should be set in terms of the global important of nutrient cycles, with reference to climate change. In this context, it would be beneficial to refer to studies that have looked at nitrogen transport more widely, and to paint the broad context as to what could happen to nitrogen flows into the ocean under rapid climate change.

We feel further that the specific aims of the paper are not clearly set out. An additional paragraph at the end of the introduction setting out the rationale and aims of the paper, and setting in a global context, would be useful.

2) Sampling, Extrapolation and Confidence Intervals: Many of the results (from "mean nitrate concentrations" in section 3.1 to final TN fluxes from the GrIS in 3.3 ) lack uncertainty values and/or confidence intervals. There is also a general lack of statistical analysis throughout the results section, in particular related to supposed trends seen in e.g. Figure 5. As this study involves extrapolation from a single year of measurement from a single, small, site, the confidence intervals will inevitably be large: this makes it all the more critical they are estimated.

3) Discharge volumes: for the calculations and extrapolations to be valid it is essential we understand the volume of glacial runoff over the melt season. The paper omits raw measurements of water volume, which would enable a better understanding of how nitrogen flux values were derived. Only a singular value of water volume arising from Leverett Glacier is provided in the paper (Table 2). This is reported as an annual measurement for 2012, despite field work occurring over the summer period only. A more comprehensive account of the measurements taken to reach the reported figure of 2.2km3 a-1, and whether this is a total actually measured or a value extrapolated from limited measurements, would hence be beneficial. Again confidence intervals would be useful here, and an idea of how much this might vary from year to year could be included in the discussion.

The paper would also benefit from expanding and enlarging the bulk discharge (output) time series (Figure 6) and presenting the continuous stage-discharge (moulin meltwater) time series (SI Figure 4) in the main article. This would enhance the interpretation of results arising from each flow measurement. Finally, the number of discharge measurements and water samples taken in the study should also be included, to improve our understanding of the methodology.

4) Sampling sites: Whilst section 2.1 adequately introduces the Leverett Glacier and its relationship with the Søndre Strømfjord, the references to Figure 1 are not well matched (NOTE: the fjord is incorrectly spelt as "Sondre Strømfjord" in text and in Figure 1). In Figure 1, the Danish name for the fjord "Søndre Strømfjord" has been replaced with "Kangerlussuaq Fjord", yet is still referred to as Sondre Strømfjord in the figure caption. Furthermore, the figure caption references "the two other major

runoff sources to the fjord. . .", which omits the Watson River. Also, the sampling site indicators are not well described: "white dot" for LG could be confused with the River locations; "red dot" refers to a red star.

We suggest that this may be resolved by changing the caption to: "Figure 1. Map showing the study area, including the location of the Leverett Glacier runoff sampling station (yellow star) and the surface sampling site (red star), together with the Watson River and the two other major runoff sources to the Søndre Strømfjord (Umivit River and Sarfartoq River)." This would require altering the map itself to rename the fjord and add the yellow star for the LG sampling station.

On a similar note, whilst the locations of the surface and bulk meltwater sampling stations are included in Figure 1, the location of the basal ice sampling is not. Not only should this be included, but more detail on the sampling site should be given in section 2.2.3. It is assumed that the margin of the glacier is used due to difficulties in accessing more pristine inner basal ice, but this is not directly stated. Also, there is no mention of the possible variation in margin basal ice and interior basal ice. These factors should be briefly discussed to assure the reader that no significant confounding factors exist – and again if they do perhaps an analysis of uncertainty could be included with confidence intervals placed on the resulting estimates.

5) Figure 5 is unclear and we believe, as presented, very difficult to interpret. We suggest several changes:

- We believe some kind of smoothing is needed, as there is clearly a lot of noise in the datasets and seeing trends through time is difficult. For example there appears to be a significant outlier in the NH4+ trend in live anaerobic – either this is real and should be commented on, or more likely it is noise and interferes with seeing the relationships. We suggest either fitting smoothed lines and making the individual points on the graphs smaller, or binning the x axis into three time ranges (say 0-300, 300-600, >600) and showing the results as means (with CI's or box/whiskers) over those periods.

- The y axes on all four sub-figures should be the same – we believe this would make the comparisons clearer

- We still believe, with these corrections, the figure would be hard to interpret. We wonder whether the use of ratios between the points would be helpful?

- The colours used here are indistinguishable in black and white. We recommend the use of symbols, e.g. triangles, circles, crosses and squares, to allow the figure to be understood in a black and white printout.

Further more minor points:

We feel that although the applicability to the whole of Greenland is reasonably well justified, it may be overstated. It is strange that in the conclusion, the "Greenland Ice Sheet" is continuously mentioned when the findings are specific to the Leverett Glacier. We would prefer if in many points through the text Greeland Ice Sheet is replaced with 'Leverett Glacier', as it is to this that the findings can be known to conclusively refer.

Figure 2 – the legend to this figure does not state what the grey periods on the left hand graphs refer to. Further, the datapoints are too large and obscure the underlying lines, e.g. in 2a)

Figure 3 – the graphs are too small, if the size could be increased (and data points shrunk) more information could be garnered from this figure. We also wonder if it might benefit from a logged x-axis – possibly both could be displayed side by side? This might allow the cluster of lower values to be more clearly observed.

Figure 4 – the line fitted here does not seem to match that in the text (which describes the intercept as 2 +- 0.6). The intercept in the graph appears to be approximately 2.5. Also given the errors on both axes we wonder if RMA/SMA regression would be more appropriate here anyway.

Page 5 lines 12-16: moulins should be defined, cryoconite debris better defined ("summer ice" not sufficient), and cryoconite itself could be better explained. A figure with

pictures could be helpful here?

Page 6 line 7 – the dye technique is not well described – we believe this is a standard technique but a reference should be provided.
* * *

---

## Referee Comment (RC1) · Anonymous Referee #1 · 8 Mar 2016

This paper aims to quantify the nitrogen sources and export from a Greenland Ice Sheet catchment and to extrapolate the export results from this catchment to the entire Greenland Ice Sheet (GrIS). The nitrogen data comprises 62 samples of bulk runoff, 39 samples of ammonium extractions from suspended sediment, 28 samples from supraglacial water, a total of 13 samples of surface ice and cryoconite water, six samples of basal ice, and a total of 13 incubation experiments. Given the logistic difficulties in obtaining this kind of data in a remote area throughout an ablation season, the nitrogen dataset is valuable and adequate to address the aim to quantify the nitrogen sources within the catchment. However, the analysis becomes more problematic with respect to cycling and export; and hence the rough extrapolation to the entire GrIS becomes very questionable.

I have some major issues with this paper:

[Figure]

1) I find it problematic that Hawkings et al. (2015) have already published some of the key data and calculations for the Leverett Glacier catchment. Hawkings et al. (2015) estimate the exchangeable ammonium flux and the DIN flux for 2012 using both discharge weighted mean (DWM) and electrical conductivity. Together with fluxes for total solutes, Si and P, these fluxes are extrapolated back in time to the years 2009, 2010 and 2011. With regards to nitrogen, Hawkings et al. (2015) find that the "annual nitrogen flux is more sensitive to changes in ice sheet water discharge than particle flux" and "inorganic nitrogen (86 % +/- 9.8 %) . . . are higher in extreme melt years than for average years. This is significant and demonstrates the potential for nutrient release by a warming climate".

It is incomprehensible to me why the Hawkings et al. (2015) paper is not referenced, and why I as a reviewer was not informed about this highly relevant publication no matter what the status of the Hawkings et al. (2015) paper was at the time of submission. In my opinion it would have been ethically correct to provide the editor and reviewers with an opportunity to assess the redundancy of parts of these papers. Why is the data and calculations presented as novel data and calculations, when they are in fact published in another paper? Both papers are written by almost the same group of authors. The paper by Hawkings et al. (2015) was accepted by Geochemical Perspectives Letters on June 19, 2015, and published on June 23, 2015, whereas this Biogeosciences paper was submitted on September 21, 2015.

As some of the key data and calculations are already published, the originality and impact of this paper are severely reduced. Clearly, many parts of the paper must be rewritten and refocused. For instance, Hawkings et al. (2015) have already discussed potential future changes in nitrogen flux (chapter 3.3). If the authors choose to rewrite the paper, then the focus should be on the novel parts of the data; i.e. the nitrogen sources. However, I am uncertain whether there is enough new data in this paper to warrant a separate publication. The rough and questionable extrapolation to the entire Greenland Ice Sheet is not sufficient.

In addition, the findings by Lawson et al. (2014) on export of nitrogen-rich organic matter should be presented in the Introduction chapter to clearly identify the gap of knowledge.

2) Another major issue is the lack of information about discharge and catchment area. Solid estimates of nitrogen concentrations, discharge measurements throughout the ablation season, and a realistic estimate of the catchment area are needed to perform the upscaling of fluxes to the entire Greenland Ice Sheet. Of these, only the nitrogen export estimates are addressed in some detail.

The discharge measurements must be described more rigorously and uncertainties in discharge should be included in total uncertainty estimates of the annual and upscaled fluxes. On P6L4-5, it is said that stage was recorded throughout the 2012 melt season (May-October), but only the first part of the runoff time series is presented in Figure 2d. Readers need to see the entire time series to get an impression of the amount of runoff after the sampling period (May 11 – July 15). Also, why does not this paper mention that the discharge from Leverett Glacier in 2012 was extreme compared to previous years (Hawkings et al., 2015)? This is rather important information, if nitrogen fluxes correlate with discharge. Why is the "runoff flux for June, July and August in 2012 from LG (2.2 km3)" (P11L13-14), when Hawkings et al. (2015) report an annual runoff flux of 2.03 km3 (their Table 1)?

It is even worse with regards to the estimated catchment area. It is postulated that it is 600 km2 (P4L4 and P13L17) without any references or a figure showing the catchment area. I tried to find out how this estimate was derived. Hawkings et al. (2015) use the same catchment area and write that "the catchment area was determined from a surface digital elevation model (Palmer et al., 2011)". Their Figure 1 shows that the entire catchment area is located well below the equilibrium line altitude for the years 2009-2012. Clearly, the use of surface digital elevation models to estimate the catchment area of relatively small ice sheet catchments constrained within the ablation area is an inappropriate method, as it does not include the upper accumulation area that

supplies ice to the lower ablation area. Assuming an equilibrium state accumulation-area ratio AAR0 between 50 % and 60 % (Dyurgerov et al., 2009), a more realistic, but still very rough, estimate of the catchment area will be between 1200 km2 and 1500 km2. The estimate of the catchment area has immense impact on the results of the upscaling and must be defined and discussed in detail. In Google Scholar, I checked some of the papers that have referenced Palmer et al. (2011) and it seems that many papers have used a catchment area of 600 km2 to calculate fluxes from the Leverett Glacier catchment. This leaves me very skeptical to many results and conclusions in a series of papers, and this is definitely not an example to follow. Unless the authors can show strong convincing arguments for using a 600 km2 catchment, I recommend that they include an estimate of the accumulation area in the total catchment area in future publications.

Minor comments:

Title: The title does not reflect the main content of the paper. The title should reflect that the main focus and data is related to the Leverett Glacier catchment. In my opinion, the rough extrapolation to the Greenland Ice Sheet is merely a simple order-of-magnitude upscaling procedure, which is too questionable to serve any real purpose without an order-of-magnitude error estimate.

Title: The paper contains very little on nitrogen cycling, i.e. flux estimates of the various pathways.

P2L19: Be specific to which "large Arctic river" the data refers to, and why this specific river is relevant in this context.

P4L16-17: Are the coordinates for Nuuk or Leverett Glacier?

P6L4-10: Insert uncertainties of the discharge determination.

P6L10-14: It is not clear to me, what the turbidity sensor was used to determine.

P6L16: Is the relationship between nitrogen concentrations and discharge constant

throughout the ablation season in other glacier catchments? If not, it may be necessary to make some kind of seasonal correction.

P6L17: Is there a potential bias due to the daily collection of water samples at 10:00? I imagine that there will be some diurnal hysteresis in a river system such as this one. Have the authors or other authors addressed this issue?

P6L18: What is the reason for only collecting water samples at 18:00 during subglacial outburst events? Is there a potential bias related to this sampling strategy? Also, I am not entirely sure what is meant by subglacial outburst events, so a definition and some information about their duration and frequency will be appreciated.

P7L2: What is the Leverett/Russell Glacier catchment? Is it just because the Leverett Glacier catchment has two names, or is it a subcatchment of the Leverett Glacier catchment? It is not shown in Figure 1 or in Hawkings et al. (2015).

P7L4-6: How did you wrap foil around 30 m x 30 m x 30 m ice blocks? I guess that there is a problem with units here.

P7L4: How many blocks were collected and where were they collected?

P7L7: It will be more informative to use cm3 rather than cm2.

P8L25: Insert Glacier after Leverett.

P9L24: What are the transect samples? If you have additional relevant data on total nitrogen, it should be included in this paper.

P10L23: At what temperature was the sediment oven-dried?

P11L5: Why were the nitrogen fluxes just calculated for the period from June to August? Why not the entire ablation season? Does this exclusion of the early and late ablation season have an effect on the upscaling?

P11L6-7: I checked the Mikkelsen and Hasholt (2013) reference to see whether it was a

valid citation in this context. The reference contains data from 2007 to 2010. It does not include anything to suggest that "LG contributed just under one half of the cumulative glacial runoff to Watson River in summer 2012". It is clearly an invalid reference. I have not time to check all references and it is not my job as a reviewer, but this makes me skeptical to the use of references in this paper. I have the impression that important relevant papers are left out, while irrelevant papers are used to support arguments, which they actually do not support.

P11L9: What is the error of the discharge weighted mean concentration method? Why is the electric conductivity method (Hawkings et al., 2015) not used in this paper?

P11L21-22: "Currently, there are no other seasonal time series of nitrogen concentrations in runoff from large Greenland outlet glaciers" – A quick Google Scholar search on Greenland+river+nitrate revealed some papers that may contain relevant data on nitrate concentrations in Greenlandic rivers. I did not check the content of these papers, so I actually do not know whether they are relevant, but the authors may find it worthwhile to do a more thorough check of the current state of knowledge on nitrogen concentrations in rivers in Greenland.

P12L1-4: I cannot follow this argument. How can the 600 km2 Leverett Glacier catchment be representative for large catchments draining the ice sheet, if it does not include a part of the accumulation area?

P12L2: Insert the maximum-minimum range.

P12L9-10: This argument assumes the same glacial history all around the ice sheet margin. On P4L21-22, the authors mention that Leverett Glacier was positioned tens of kilometers further inland during the Holocene Thermal Maximum. Were all outlet glaciers from the Greenland Ice Sheet located further inland during the Holocene Thermal Maximum?

P12L10: Is it relevant for extrapolation to tidewater glaciers that Leverett Glacier is a

land-terminating glacier?

P12L10: How much nitrogen derives from the local bedrock? How much nitrogen is contained in other types of bedrock beneath the GrIS?

P12L13-16: This is exactly the kind of arguments that makes me concerned. If an incorrect catchment area was used in previous papers, then there are no good reasons to continue using it. This is not a valid argument for considering the Leverett Glacier catchment as representative for all large GrIS catchments.

P13L23: Is a termination of nitrogen fluxes in late July/early August supported by data from other catchments?

P15L3-4: So there is a bias caused by the collection of water samples at 18:00?

P15L24-25: There is an issue with wrong units here.

P16L4: Insert error estimates and discuss these.

P16L11-12: Is this correlation linear or exponential?

P16L12-27: This discussion on potential future changes in largely a repetition of Hawkings et al. (2015). The discussion in this paper needs to be different.

Table 1: What is meant by "Moulins (same period)"?

Table 2: What is meant by seasonal fluxes in the caption? The fluxes from the Leverett Glacier catchment should use the unit tons a-1.

References:

Dyurgerov, M., Meier, M.F. and Bahr, D.B. 2009. A new index of glacier area change: a tool for glacier monitoring. Journal of Glaciology, 55, 710-716.

Hawkings, J.R., Wadham, J.L., Tranter, M. et al. 2015. The effect of warming climate on nutrient and solute export from the Greenland Ice Sheet. Geochemical Perspectives Letters, 1, 94-104.

Lawson, E.C., Bhatia, M.P., Wadham, J.L. and Kujawinski, E.B. 2014. Continuous summer export of nitrogen-rich organic matter from the Greenland Ice Sheet inferred by ultrahigh resolution mass spectrometry. Environmental Science and Technology, 48, 14248-14257.

Palmer, S., Shepherd, A., Nienow, P. and Joughin, I. 2011. Seasonal speedup of the Greenland Ice Sheet linked to routing of surface water. Earth and Planetary Science Letters, 302, 423-428.

---

## Referee Comment (RC2) · Anonymous Referee #2 · 15 Mar 2016

This paper presents and interesting dataset regarding nitrogen contents in meltwater and glacial ice within the Greenland Ice Sheet – more specifically the Leverett Glacier. The dataset is very interesting and while I don't have a problem with the authors' extrapolation of their results to the entire Greenland ice sheet I think many details need to be clarified in their paper. The dataset should be viewed in a far more quantitative way (the current manuscript is almost devoid of statistics) and the discussion of nitrogen cycling should be increased significantly. There is almost no mention of mechanisms behind the additional sources of nitrogen, the experiments they did are poorly reported (though could possibly provide interesting information). If the authors are interested in an observational paper that reports interesting concentrations and calculates nitrogen fluxes that could be acceptable but the title of the paper should then be changed to reflect the lack of discussion on processing.

[Figure]

The extrapolation of results As stated – I don't have a problem with the extrapolation but I think that the reader needs to know more so that they can put it into context. For example – what percentage of the GIC drains to the Leverett Glacier? How do they determine this? This "watershed" calculation/determination is important because of the focus they put on comparing their results with that of the PARTNERS/GRO project on the Arctic rivers. The watershed area of the Lena, Ob, etc. is determined by topography how is the ice sheet drainage area determined? How good is this approximation? For example in the Arctic rivers they acknowledge that the watersheds as determined by topography may underestimate groundwater or permafrost sources given the terrain and unknown flow paths. Also – the authors state that it is ok to extrapolate because a large portion of the GIC is underlain by similar bedrock geology. Can the authors be more quantitative in this statement? What is the percentage?

Sources of N & discussion of processing The authors need to expand their discussion of N cycling in glacial environments. They continuously cite one study – however there is ample evidence of the processes that they are invoking from the alpine glacial community (e.g. Baron, Saros, Brooks & Williams, Clow in Rocky Mountain systems alone) or even more evidence within the snowpack studies.

In addition, there was significant explanation provided for the setup of the incubation experiments (though there should be further explanation to the choice of ice sampling location- presumably it was due to access?) and then the results of these experiments are barely mentioned and poorly presented. Figure 5 needs significant improvement. Simple things like making all the y axes the same scale would help the reader interpret your findings. They are still too busy though. What would be extremely helpful to is show changes in the various N species over time – maybe even with a bar chart (+ gain, - loss) that represents the net aerobic and net anaerobic processing for each species (NO3-, DON, NH4+ – I don't think TDN is needed). This would allow you to better track how the species are changing, quantify rate changes, do statistical tests on whether the "live" experiments were different than the water only ones. Overall this

section needs significant improvement and then contextualization within the literature. What do these experimental fluxes represent? Is there any way to relate them to the N yields reported?

In terms of the discussion of source (relationship between moulin waters and bulk runoff – Figure 4), the authors also need to be more quantitative. The authors have enough solute data to do some sort of endmember analysis (or at the very least repeat the basic approach used in Figure 4 with other solutes that should also be unique to the meltwater). Figure 3 could be improved by using a ternary diagram instead to try and separate out sources visually. The authors show sulfate concentrations in Figure 6 but don't sufficiently use this information within the text. Sulfate is likely a weathering product UNLESS there is also high levels of sulfate in the glacial ice (which is possible given industrial emissions). A discussion regarding the possible sources of sulfate should be added.

The calculation of nitrogen and sediment fluxes – Why was a volume weighted average used instead of creating a relationship between discharge and concentration and then estimating the N concentrations for each daily Q value? It appears that there are discharge estimates ever 5 or 10 minutes and daily nitrogen values – surely that is enough data to figure out these relationships and therefore do a better job of quantifying (and discussing error) on the N fluxes.

Specific editorial notes: Abstract, line 9: remove "and" Pg 4, line 20-21: change to "in the last few thousand years" Pg 15, lines 24-25: typically these yields are referred to in "m a-1" or "m yr-1" (i.e. the units are "reduced") Pg 16, line 7: change "to" to "than"

---

## Author Comment (AC1) · 19 May 2016

We thank the reviewers for their comprehensive set of comments on our manuscript. We respond to their individual commentary in the following paragraphs.

Referee 1 (484-RC1)

Major comments. 1. "I find it problematic that Hawkings et al. (2015) have already published some of the key data and calculations for the Leverett Glacier catchment... Why is the data and calculations presented as novel data and calculations, when they are in fact published in another paper?"

Hawkings et al (2015) present a very small portion of the data featured in the BD manuscript. The only data Hawkings et al (2015) report are dissolved inorganic nitro-
gen fluxes (DIN=NH4+ + NO3-) and SS-NH4+ in subglacial runoff, the former calculated from a regression between electrical conductivity and DIN for three melt seasons. They exclude DON. They also do not report either a) concentrations of any nitrogen species in snow, ice surface waters, basal ice and b) any data that would enable ice sheet nitrogen cycling processes to be inferred (e.g. incubation experiments, comparison of surface and basal waters) and c) they do attempt to scale their fluxes to the entire ice sheet. The discussion of DIN fluxes in Hawkings et al comprised approx. 5 lines of text in the entire manuscript. We have now referred to this paper. We disagree with the reviewer that there is limited novel data in the BC manuscript. Please see the paragraph beginning Line 66. In line with comments from other reviewers we have also expanded the discussion of nitrogen sources and cycling (Pages 12-14).

2. Another major issue is the lack of information about discharge and catchment area. The discharge measurements must be described more rigorously and uncertainties in discharge should be included in total uncertainty estimates of the annual and upscaled fluxes.

The reason that the discharge and catchment areas are not described more fully is that these are published elsewhere, and we refer to these papers (Line 126 and 269), which is normal practice. The catchment area was calculated in Cowton et al (2012) and the 2012 discharge data has previously been published in (Hawkings et al., 2014;Hawkings et al., 2015;Tedstone et al., 2013). The error in discharge measurements is +/-10% (see (Tedstone et al., 2013). We have now made clearer reference to this work. The same is true of the MAR regional climate model used to constrain the Greenland Ice Sheet runoff terms: for such models typical error terms on surface mass balance modeling are of the order of 10% (Vernon et al., 2013). We now quote these error terms in the text and note that any nutrient fluxes calculated from such data have an associated 10% uncertainty (see Line 269 and 295).

- On P6L4-5, it is said that stage was recorded throughout the 2012 melt season (May-October), but only the first part of the runoff time series is presented in Figure 2d.

[Figure]

We have now added a separate plot in supplementary information showing the full seasonal discharge cycle (Supplementary Figure 6), referred to on Line 264.

- Also, why does not this paper mention that the discharge from Leverett Glacier in 2012 was extreme compared to previous years (Hawkings et al., 2015)?

This omission was unintentional, but the 2012 melt season has been documented many times elsewhere (e.g. (Hawkings et al., 2015;Tedesco et al., 2013). We now also make reference to this point (Line 255).

- Why is the "runoff flux for June, July and August in 2012 from LG (2.2 km3)" (P11L13-14), when Hawkings et al. (2015) report an annual runoff flux of 2.03 km3 (their Table 1)?

As stated in the Hawkings et al (2015), the discharge data presented in the GPL paper was normalised to 17th August in all the comparison years (clearly noted in the Table 1 and Figure 2 subscript) because of differences in the length of river monitoring for each of the years, so the estimate is smaller for 2012 in this case. Our previous work on the 2012 melt season, like the BD manuscript, used 2.2 km3 a-1 e.g. (Hawkings et al., 2014;Hawkings et al., 2016).

- Catchment area. It is postulated that it is 600 km2 (P4L4 and P13L17) without any references or a figure showing the catchment area.

Full details of how the catchment area was determined are included in many other papers, most notably (Cowton et al., 2012), which is now referenced in the manuscript. This is a modeled estimate of catchment area for the runoff measured at Leverett Glacier portal, and is essentially the area of catchment required to explain the volume and timing of runoff at the glacier margin. By "catchment area", we are referring to the active hydrological catchment contributing runoff measured at the Leverett Glacier portal (Line 323), consistent with our previous work. The value of 600 km2 has a generous estimate of +/- 25% uncertainty, as discussed in (Cowton et al., 2012). We also now

note this in the text (Line 324). In response to the reviewer's comments about including the accumulation area, we note that Leverett Glacier is bordered by large glacier systems to both the north and south. Indeed, modeling in (Cowton et al., 2012) generates a very similar modeled catchment area in contrasting melt years, despite evidence of melt to much higher elevations during high melt years (e.g. the high melt year of 2010). This implies that the catchment is topographically defined, with meltwater from further inland draining to the north or south. Because of this, it is inappropriate to use a larger catchment area that penetrates further into the accumulation zone. (Palmer et al., 2011) did calculate (using a surface DEM routing model) a much larger catchment (1200 km2), but this included the neighbouring Russell Glacier catchment and water routing could only be constrained by surface topography. For our purposes, we consider the 600 km2 catchment the most applicable because it applies only to Leverett Glacier and does not consider higher elevation zones of melting, where meltwater was routed to larger catchments to the north and south. We acknowledge that defining catchment areas in an ice sheet setting will always be fraught with uncertainty, but believe that we have used the best approach for our purposes.

Minor comments

1. Title: The title does not reflect the main content of the paper. The title should reflect that the main focus and data is related to the Leverett Glacier catchment. In my opinion, the rough extrapolation to the Greenland Ice Sheet is merely a simple order-of-magnitude upscaling procedure, which is too questionable to serve any real purpose without an order-of-magnitude error estimate.

We disagree with this comment. This is the first and only study on nitrogen cycling on the Greenland Ice Sheet, and examines the sources of nitrogen, how nitrogen is cycled and fluxes from the catchment and ice sheet (as the title suggests). To address the issues of uncertainty, we now provide minimum, mean and maximum estimates of nitrogen fluxes from both Leverett Glacier and the Greenland Ice Sheet (Line 292), employing a wider envelope of nitrogen concentrations in runoff to calculate fluxes. We

also note an error term of ±27% on both flux estimates for Leverett catchment, which incorporate both the combined discharge (10%) uncertainty and catchment error (25%) uncertainty (Line 271). For the Greenland Ice Sheet, the modeled meltwater runoff has an error term of 10% (Line 296)(Vernon et al., 2013). While these considerations do produce large uncertainty terms in flux estimates (which we already acknowledged – see Line 297), they provide the first values to date that can be compared to other nitrogen sources in the Arctic region. Hence, we believe that this is still a valuable exercise, and at present the best estimation of ice sheet nitrogen fluxes.

2. Title: The paper contains very little on nitrogen cycling, i.e. flux estimates of the various pathways.

We have now included further discussion of nitrogen cycling (see Pages 13 and 14).

3. P2L19: Be specific to which "large Arctic river" the data refers to, and why this specific river is relevant in this context.

Amended – it is specific to each of the largest Arctic rivers

4. P4L16-17: Are the coordinates for Nuuk or Leverett Glacier?

These coordinates are for Leverett Glacier.

5. P6L4-10: Insert uncertainties of the discharge determination.

We have now inserted uncertainties. Please see Line 270, 296

6. P6L10-14: It is not clear to me, what the turbidity sensor was used to determine.

The turbidity sensor was used to determine turbidity, which is a proxy for suspended sediment concentration. We have now added a few words by way of explanation (see Line 132)

7. P6L16: Is the relationship between nitrogen concentrations and discharge constant throughout the ablation season in other glacier catchments? If not, it may be necessary

to make some kind of seasonal correction.

The relationship between nitrogen concentrations and discharge is usually an inverse non-linear relationship (see Hawkings et al 2015), since TDN (particularly nitrate which comprises a large proportion of TDN) concentrations normally decline exponentially through the season as the snowpack retreats. This is commonly reported on other glacier systems (Hodson et al., 2010). We have not applied any seasonal correction (we presume the reviewer means to our reported mean nitrogen concentrations) since we calculate discharge-weighted mean concentrations of nitrogen species, which biases our mean concentration towards mid-seasonal discharge values (which are low).

8. P6L17: Is there a potential bias due to the daily collection of water samples at 10:00? I imagine that there will be some diurnal hysteresis in a river system such as this one. Have the authors or other authors addressed this issue?

Because of the high lag times present in this large catchment and 24 hours daylight, there is only small diurnal variation between daily samples collected in the morning versus the evening. As stated in the manuscript, we also collected samples at 18.00. There is some difference (not statistically significant) between the concentrations of N species in 10.00 versus 18.00 samples. During the peak melt period, concentrations are 5-10% higher in the afternoon. This would mean that our flux estimates are conservative since more samples were collected at 10.00 than 18.00. It is hard to discern whether this difference reflects diurnal variation since the occurrence of outburst events also have an impact on solute concentrations in runoff.

9. P6L18: What is the reason for only collecting water samples at 18:00 during subglacial outburst events? Is there a potential bias related to this sampling strategy? Also, I am not entirely sure what is meant by subglacial outburst events, so a definition and some information about their duration and frequency will be appreciated.

We collected additional samples at 18.00 during outburst events since the discharge is varying rapidly at these times, and we wished to capture the dynamics of solute and

nutrient release during these events. This is now noted in the text (Line 141).

10. P7L2: What is the Leverett/Russell Glacier catchment? Is it just because the Leverett Glacier catchment has two names, or is it a subcatchment of the Leverett Glacier catchment? It is not shown in Figure 1 or in Hawkings et al. (2015).

Russell Glacier is the small glacier located to the north of Leverett Glacier. We have now labeled this on Figure 1.

11. P7L4-6: How did you wrap foil around 30 m x 30 m x 30 m ice blocks? I guess that there is a problem with units here.

We used large and multiple sheets of pre-combusted foil. This is now included in the text (Line 156). We have also change the units to cm.

12. P7L4: How many blocks were collected and where were they collected?

The location of the blocks is now included in Figure 1. >10 blocks were collected but analyses in this paper referred to those conducted upon a sub-set of these.

13. P7L7: It will be more informative to use cm3 rather than cm2.

We now note the dimensions of these chunks of ice. See Line 157.

14. P8L25: Insert Glacier after Leverett.

Amended – Line 200.

15. P9L24: What are the transect samples? If you have additional relevant data on total nitrogen, it should be included in this paper.

These were included in an earlier version of the paper, all text related to them has now been deleted. We thank the reviewer for pointing this out.

16. P10L23: At what temperature was the sediment oven-dried?

Overnight at 40°C. This is now mentioned in the text (Line 246).

17. P11L5: Why were the nitrogen fluxes just calculated for the period from June to August? Why not the entire ablation season? Does this exclusion of the early and late ablation season have an effect on the upscaling?

The 2.2 km3 refers to the entire monitoring period (now corrected in the text, Lines 254 and 263).

18. P11L6-7: I checked the Mikkelsen and Hasholt (2013) reference to see whether it was a valid citation in this context. The reference contains data from 2007 to 2010. It does not include anything to suggest that "LG contributed just under one half of the cumulative glacial runoff to Watson River in summer 2012". It is clearly an invalid reference. I have not time to check all references and it is not my job as a reviewer, but this makes me skeptical to the use of references in this paper. I have the impression that important relevant papers are left out, while irrelevant papers are used to support arguments, which they actually do not support.

We have now deleted this section, as it is not strictly relevant to the manuscript.

19. P11L9: What is the error of the discharge weighted mean concentration method? Why is the electric conductivity method (Hawkings et al., 2015) not used in this paper?

There are various ways we could calculate the total nitrogen fluxes from Leverett Glacier catchment – a discharge weighted mean method (which is commonly used), via the regression relationship between nitrogen concentrations and discharge and via the method employed in Hawkings et al (2015) which was used because nitrogen concentration data was not available in all years of study and hence, this was calculated from electrical conductivity. A discussion of the difference between the first and last of these methods is provided in Hawkings et al (2015) which states a value of 7% difference. This is a small error term and exceeded by other error terms (e.g. discharge). It would not have been appropriate to use either the Q/solute relationship of EC/solute relationship for all nitrogen species since DON is not correlated to EC or discharge. Hence, we continue to use the discharge weighted mean concentration method, which

is widely applied.

20. P11L21-22: "Currently, there are no other seasonal time series of nitrogen concentrations in runoff from large Greenland outlet glaciers" – A quick Google Scholar search on Greenland+river+nitrate revealed some papers that may contain relevant data on nitrate concentrations in Greenlandic rivers. I did not check the content of these papers, so I actually do not know whether they are relevant, but the authors may find it worthwhile to do a more thorough check of the current state of knowledge on nitrogen concentrations in rivers in Greenland.

We do not know which papers the reviewer refers to here, as these are not referenced. To the best of our knowledge and belief ours is the first dataset on nitrogen concentrations in runoff from a large Greenlandic glaciated catchment.

21. P12L1-4: I cannot follow this argument. How can the 600 km2 Leverett Glacier catchment be representative for large catchments draining the ice sheet, if it does not include a part of the accumulation area?

The accumulation area does not supply runoff to the glacier margin, please see our discussion of the catchment area under point 2 of "Major Comments".

22. P12L2: Insert the maximum-minimum range.

We have now included the altitudinal range (250-1510 m asl). This is now mentioned in the text (Line 278).

23. P12L9-10: This argument assumes the same glacial history all around the ice sheet margin. On P4L21-22, the authors mention that Leverett Glacier was positioned tens of kilometers further inland during the Holocene Thermal Maximum. Were all outlet glaciers from the Greenland Ice Sheet located further inland during the Holocene Thermal Maximum?

The retreat history of the Greenland margin is not well constrained. The argument made here only assumes that sedimentary ecosystems are widespread beneath the

ice sheet.

24. P12L10: Is it relevant for extrapolation to tidewater glaciers that Leverett Glacier is a land-terminating glacier?

It would be impossible to constrain nitrogen fluxes from a tidewater glacier. However, the nitrogen sources in both systems (ice, snowmelt, glacial ecosystems) are likely to be the same. Hence, we believe that this is a valid extrapolation.

25. P12L10: How much nitrogen derives from the local bedrock? How much nitrogen is contained in other types of bedrock beneath the GrIS?

The nitrogen content of the bedrock (largely gneiss) is likely to be low (e.g. <10 mg N kg-1 cited for Greenland gneiss (Holloway and Dahlgren, 2002). We found that it was below the detection limit of combustion analysis methods. We have now added some mention of this in the paper – see Line 399.

26. P12L13-16: This is exactly the kind of arguments that makes me concerned. If an incorrect catchment area was used in previous papers, then there are no good reasons to continue using it. This is not a valid argument for considering the Leverett Glacier catchment as representative for all large GrIS catchments.

Please see our previous discussion of catchment area (Major Issues, point 2). The wrong catchment area has not been used.

27. P13L23: Is a termination of nitrogen fluxes in late July/early August supported by data from other catchments?

There is no data from other catchments in Greenland. However, we still see significant nitrogen concentrations in moulin waters in early August (6 uM) and the sources of this nitrogen (icemelt and supraglacial ecosystems) will continue to be active sources while the catchment remains hydrologically active. We do not argue for a termination of these fluxes in August, but rather that they are likely to continue while runoff is sustained at the margin.

[Figure]

28. P15L3-4: So there is a bias caused by the collection of water samples at 18:00?

There is no significant bias introduced by using these samples, and they were collected in addition to the 10.00 samples (now noted in the text) and aimed to capture the dynamics of solute release during subglacial outburst events.

29. P15L24-25: There is an issue with wrong units here.

This is corrected now to m3 km-2 a-1

30. P16L4: Insert error estimates and discuss these.

We have now introduced upper and lower bounds to these values, also mentioning the error introduced from the runoff errors (Line 420).

31. P16L11-12: Is this correlation linear or exponential?

This correlation is exponential. We now note this in the text (see Line 429).

32. P16L12-27: This discussion on potential future changes in largely a repetition of Hawkings et al. (2015). The discussion in this paper needs to be different.

We disagree with this statement. There is minimal text in Hawkings et al (2015) that specifically discuss Greenland nitrogen fluxes in a warming climate. The only common feature is the hypothesis that nitrogen fluxes will rise in warmer melt years and that the importance of dissolved fluxes as a proportion of the total load will mean heightened sensitivity, but very little detail is given on this in Hawkings et al (2015). We now refer to this paper in the discussion, and build upon it.

33. Table 1: What is meant by "Moulins (same period)"?

We have changed this to "same time period" in the table. We excluded the moulin samples for which there were not contiguous runoff samples for this data summary.

34. Table 2: What is meant by seasonal fluxes in the caption? The fluxes from the Leverett Glacier catchment should use the unit tons a-1.

This table has been replaced by two new tables. We have now changed the units for Leverett Glacier to ton a-1.

Referee 2 (484-SC1)

This review (prepared by a group of people) is largely supportive. They note the novelty of the dataset and the soundness of the methodology. We respond to their substantive comments as follows:

1. Introduction: While the introduction to this paper is adequate, it is very short and we have identified some potential improvements that could be made. We feel the introduction is somewhat brief and lacks explanation for why the study site was chosen nor the background behind the study and the methods used. They recommend the following revisions:

- Note of why Leverett Glacier is representative

There is substantial text on this in (Line 265 onwards). We have also now added a sentence in the introduction section (Line 73). We do not include the % of glacier cover that this catchment accounts for – it will be a very small number, like most catchments in Greenland. This is a very different situation to Arctic rivers, where the total runoff is dominated by 6 large systems (Holmes et al., 2012). In Greenland, many glacial outlets deliver runoff from the ice sheet to the ocean, where even the very large catchments only represent <5% of the total ice sheet area. The argument of how representative a catchment is largely rests on a) the bedrock geology, b) size and related to this, whether it drains inland meltwaters. Leverett Glacier satisfies both of these criteria as stated in Line 277 onwards.

- You state "there is comparatively little data on nitrogen sources and cycling on the Greenland Ice Sheet, which is likely to be important as a nutrient source to downstream fjord and marine ecosystems." While this provides some background to the specifics of the Greenland ecosystem, this should be set in terms of the global important of nutrient

cycles, with reference to climate change. In this context, it would be beneficial to refer to studies that have looked at nitrogen transport more widely.

We have endeavored to keep the introductory section concise, and relevant to Greenland and its bordering ocean masses. We would be more than happy to include wider information if the reviewers can give us an indication of what they think is missing in terms of wider global cycles. It's worth noting that the export of nitrogen from Greenland is very unlikely to have more than a local to regional effect.

- We feel further that the specific aims of the paper are not clearly set out. An additional paragraph at the end of the introduction setting out the rationale and aims of the paper, and setting in a global context, would be useful

We have now added a sentence that explains the aim of the manuscript (see Line 66).

2. Sampling, Extrapolation and Confidence Intervals: Many of the results (from "mean nitrate concentrations" in section 3.1 to final TN fluxes from the GrIS in 3.3 ) lack uncertainty values and/or confidence intervals. There is also a general lack of statistical analysis throughout the results section, in particular related to supposed trends seen in e.g. Figure 5. As this study involves extrapolation from a single year of measurement from a single, small, site, the confidence intervals will inevitably be large: this makes it all the more critical they are estimated.

We thank the reviewers for raising this point, and have endeavored to give a better estimate of the uncertainty in nitrogen fluxes from the catchment and ice sheet. While there are inevitable errors in the discharge data and catchment area (now reported in the text – see Line 270), probably the greatest uncertainty for the ice sheet extrapolation and perhaps between years for Leverett catchment is the concentration of nitrogen species in runoff. To give a maximum estimate of how this uncertainty might propagate into overall flux calculations, we now employ minimum, mean (discharge weighted) and maximum concentrations for nitrogen species in order to calculate a potential envelope of nitrogen fluxes both from Leverett catchment and the ice sheet. These values are reported in Tables 2 and 3 and also in the text. We hope that this satisfied the reviewers' query.

3. Discharge volumes: for the calculations and extrapolations to be valid it is essential we understand the volume of glacial runoff over the melt season. The paper omits raw measurements of water volume, which would enable a better understanding of how nitrogen flux values were derived. Only a singular value of water volume arising from Leverett Glacier is provided in the paper (Table 2). This is reported as an annual measurement for 2012, despite field work occurring over the summer period only. A more comprehensive account of the measurements taken to reach the reported figure of 2.2km3 a-1, and whether this is a total actually measured or a value extrapolated from limited measurements, would hence be beneficial. Again confidence intervals would be useful here, and an idea of how much this might vary from year to year could be included in the discussion

We have now included a plot of the seasonal discharge hydrograph in the Supplementary information (Supp. Figure 5) so that the reader may gain an appreciation for the raw data from which the total water flux was derived. Measurements were taken every 5-10 minutes (see Methods Section) and we interpolated between these data points to derive a total seasonal flux. We hope that this is helpful. The proglacial discharge drops to virtually zero by mid September, which accounts for the reported 2.2 km3. We also now report the uncertainty in these discharge data which is estimated at 10% (Tedstone et al., 2013) (Line 270).

4. The paper would also benefit from expanding and enlarging the bulk discharge (output) time series (Figure 6) and presenting the continuous stage-discharge (moulin meltwater) time series (SI Figure 4) in the main article

The bulk discharge time series in Figure 6 is deliberately truncated as this plot is intended to give a zoomed in view of trends in nitrogen concentrations during outburst events. Hence, we retain the scale as it stands. We do now include the full discharge

time series in the supplementary information (Supp. Figure 5). This is already published in (Hawkings et al., 2015;Tedstone et al., 2013)We do not report moulin discharge and nitrogen flux time series in the main manuscript, hence we judge that it would be confusing to add the moulin water flux time series here. This figure has been retained in the supplementary information.

5. Finally, the number of discharge measurements and water samples taken in the study should also be included, to improve our understanding of the methodology.

This information is already present in the Methods Section and in Table 1.

6. Sampling sites: Whilst section 2.1 adequately introduces the Leverett Glacier and its relationship with the Søndre Strømfjord, the references to Figure 1 are not well matched (NOTE: the fjord is incorrectly spelt as "Sondre Strømfjord" in text and in Figure 1). In Figure 1, the Danish name for the fjord "Søndre Strømfjord" has been replaced with "Kangerlussuaq Fjord", yet is still referred to as Sondre Strømfjord in the figure caption. Furthermore, the figure caption references "the two other major runoff sources to the fjord: : :", which omits the Watson River. Also, the sampling site indicators are not well described: "white dot" for LG could be confused with the River locations; "red dot" refers to a red star…... On a similar note, whilst the locations of the surface and bulk meltwater sampling stations are included in Figure 1, the location of the basal ice sampling is not.

We have now corrected the name of the fjord in the text and the figure. We have also corrected the caption and sampling site indicators.

7. Basal ice sampling - more detail on the sampling site should be given in section 2.2.3. It is assumed that the margin of the glacier is used due to difficulties in accessing more pristine inner basal ice, but this is not directly stated. Also, there is no mention of the possible variation in margin basal ice and interior basal ice. These factors should be briefly discussed to assure the reader that no significant confounding factors exist – and again if they do perhaps an analysis of uncertainty could be included with confidence

intervals placed on the resulting estimates.

We already stated that we sampled basal ice from the margin. Basal ice has not been sampled from the interior of this catchment, and so we cannot add any information on this. Generally basal ice can only be sampled from marginal outcrops or via ice coring, which is expensive and has not been conducted in this region. Insufficient data exists on the variation in basal ice chemistry to be able to assess the uncertainty in basal ice nitrogen concentrations.

8. Figure 5 - We believe some kind of smoothing is needed, as there is clearly a lot of noise in the datasets and seeing trends through time is difficult. For example there appears to be a significant outlier in the NH4+ trend in live anaerobic – either this is real and should be commented on, or more likely it is noise and interferes with seeing the relationships. We suggest either fitting smoothed lines and making the individual points on the graphs smaller, or binning the x axis into three time ranges (say 0-300, 300-600, >600) and showing the results as means (with CI's or box/whiskers) over those periods.

Since the incubations were sampled roughly every 100 days and there are only 6 or so data points, we do not consider smoothing a robust approach in this case. Neither do we support binning such a small number of time points, which will disguise the variability (which we believe is natural). We do take on board these comments, however, and have amended the Figure substantially.

• The y axes on all four sub-figures should be the same – we believe this would make the comparisons clearer

Amended.

• We still believe, with these corrections, the figure would be hard to interpret. We wonder whether the use of ratios between the points would be helpful?

We take on board this point, and have made some changes to the figures in order to

improve clarity.

• The colours used here are indistinguishable in black and white. We recommend the use of symbols, e.g. triangles, circles, crosses and squares, to allow the figure to be understood in a black and white printout

We have now improved this and changed the figure to black and white.

9. We feel that although the applicability to the whole of Greenland is reasonably well justified, it may be overstated. It is strange that in the conclusion, the "Greenland Ice Sheet" is continuously mentioned when the findings are specific to the Leverett Glacier. We would prefer if in many points through the text Greeland Ice Sheet is replaced with 'Leverett Glacier', as it is to this that the findings can be known to conclusively refer.

We have rephrased the conclusion section in acknowledgement of this point.

10. Figure 2 – the legend to this figure does not state what the grey periods on the left hand graphs refer to. Further, the datapoints are too large and obscure the underlying lines, e.g. in 2a)

We have now included an explanation for the grey bars, which indicate subglacial outburst events. We have also reduced the size of the data points in a) and b) and changed the scale to reduce overlap between data series.

11. Figure 3 – the graphs are too small, if the size could be increased (and data points shrunk) more information could be garnered from this figure. We also wonder if it might benefit from a logged x-axis – possibly both could be displayed side by side? This might allow the cluster of lower values to be more clearly observed.

The size of the graphs is dictated by the typesetting of the manuscript. We tried a log scale but this did not improve the clarity of the plot. Instead we have added an inset where the scale zooms into the cluster of data points at low concentrations. We hope that this improves the clarity of the Figure.

12. Figure 4 – the line fitted here does not seem to match that in the text (which describes the intercept as 2 +- 0.6). The intercept in the graph appears to be approximately 2.5.

Intercept amended.

13. Page 5 lines 12-16: moulins should be defined, cryoconite debris better defined ("summer ice" not sufficient), and cryoconite itself could be better explained. A figure with pictures could be helpful here?

We believe that these are standard terms in glaciological work, and are already defined sufficiently in the existing manuscript, and papers to which we refer in the text. For example:

"Cryoconite holes are water-filled cylindrical melt holes, formed by radiation heating of surface sediment and subsequent melting (Podgorny and Grenfell, 1996). The debris in the base of these holes is termed "cryoconite" which may become distributed over the glacier surface during melt out of cryoconite holes in summer"

"ice containing dispersed cryoconite debris (referred to here as "summer ice")"

"Here, samples of meltwater descending to the ice sheet bed via a large moulin were collected from the streams feeding the moulin between 5th May and 9th August (Day 129 and 222)"

14. Page 6 line 7 – the dye technique is not well described – we believe this is a standard technique but a reference should be provided

This is a standard method. We now provide a reference to previous work using this dye tracing for information (Line 130).

Referee 3 - 484 – RC2

This is a supportive review, claiming that the work is novel and interesting. They endorse the approach taken, but ask for further discussion of nitrogen sources.

1. The extrapolation of results As stated – I don't have a problem with the extrapolation but I think that the reader needs to know more so that they can put it into context. For example – what percentage of the GIC drains to the Leverett Glacier? How do they determine this?

We haven't included the % of glacier cover that this catchment accounts for – it is a very small number (0.02% of the total ice sheet), like most catchments in Greenland. This is a very different situation to Arctic rivers, where the total runoff is dominated by 6 large systems. In Greenland, many glacial outlets deliver runoff from the ice sheet to the ocean, where even the very large catchments only represent <5% of the total ice sheet area.

2. The watershed area of the Lena, Ob, etc. is determined by topography how is the ice sheet drainage area determined? How good is this approximation?

We do not employ the total area of the ice sheet for any calculations. The drainage basin of Leverett catchment is a modeled estimate, calculated as the area of catchment required to explain the volume of Leverett margin using a degree day approach (Cowton et al., 2012).

3. The authors state that it is ok to extrapolate because a large portion of the GIC is underlain by similar bedrock geology. Can the authors be more quantitative in this statement? What is the percentage?

It is difficult to quantify exactly how much of the GrIS comprises the type of bedrock found at Leveret Glacier because 85% of the landmass lies beneath thick ice cover. Those that have studied the geology of Greenland are hesitant to put a figure on the proportion of the land mass which comprises Archaen rocks similar to those seen at Leverett Glacier, and we concur with this. For example, (Kalsbeek and Taylor, 1984) state that Archaen rocks (gneiss and granite) in Greenland have a wide distribution, including the Archean rocks of southern Greenland and modified Archaen rocks to the north and south (see diagram below). Hence, the vast proportion of the bedrock

in Greenland comprises granite and gneiss (Kalsbeek and Taylor, 1984). We now reference this paper in our Site Description section (Line 89).

4. Sources of N & discussion of processing The authors need to expand their discussion of N cycling in glacial environments. They continuously cite one study – however there is ample evidence of the processes that they are invoking from the alpine glacial community (e.g. Baron, Saros, Brooks & Williams, Clow in Rocky Mountain systems alone) or even more evidence within the snowpack studies.

We have now extended the discussion section of N sources and cycling and reference a wider body of work (Pages 12-14.)

5. Basal ice - there should be further explanation to the choice of ice sampling location-presumably it was due to access?)

This was due to access and the presence of visible debris-rich basal ice outcrops in this area. We now mention this (Line 153).

6. Incubation experiments (Figure 5) The results of these experiments are barely mentioned and poorly presented

We have tried to improve the presentation of these data (see Figure 5). We have also added a more lengthy discussion of these data (See Page 12-14)

7. This section needs significant improvement and then contextualization within the literature. What do these experimental fluxes represent? Is there any way to relate them to the N yields reported?

We have now extended this section. The experimental results simply indicate which nitrogen cycling processes are likely to operate in subglacial environments. We hesitate to convert these to yields since the thickness of subglacial sediment beneath the ice is highly uncertain.

8. In terms of the discussion of source (relationship between moulin waters and bulk

runoff – Figure 4), the authors also need to be more quantitative. The authors have enough solute data to do some sort of endmember analysis (or at the very least repeat the basic approach used in Figure 4 with other solutes that should also be unique to the meltwater). Figure 3 could be improved by using a ternary diagram instead to try and separate out sources visually.

We plotted similar relationships for all nitrogen species following the practice in Figure 4, and the only significant relationship applied for Total Dissolved Nitrogen. We could have repeated the analysis for other solutes, but in general concentrations of crustal species in surface meltwaters are present in low to zero concentrations and these species derive largely from the ice sheet bed. Hence, we didn't feel it worthwhile to carry this out. Following comments from another reviewer, we have also revised Figure 3 to include an inset, which enables the cluster of low-concentration data points to be more easily viewed.

9. The authors show sulfate concentrations in Figure 6 but don't sufficiently use this information within the text. Sulfate is likely a weathering product UNLESS there is also high levels of sulfate in the glacial ice (which is possible given industrial emissions). A discussion regarding the possible sources of sulfate should be added.

We now discuss the sulphate time series in the text (Line 373).

References: Cowton, T., Nienow, P., Bartholomew, I., Sole, A., and Mair, D.: Rapid erosion beneath the Greenland ice sheet, Geology, 40, 343-346, Doi 10.1130/G32687.1, 2012. Hawkings, J., Wadham, J. L., Tranter, M., Raiswell, R., Benning, L. G., Statham, P. J., Tedstone, A., and Nienow, P.: Ice sheets as a significant source of highly reactive nanoparticulate iron to the oceans, Nature Communications, 5, doi:10.1038/ncomms4929, 2014. Hawkings, J., Wadham, J. L., Tranter, M., Telling, J., Bagshaw, E. A., Beaton, A., Simmons, S. L., Tedstone, A., and Nienow, P. W.: The Greenland Ice Sheet as a hot spot of phosphorus weathering and export in the Arctic, Global Biogeochem Cy, 30, 191-210, 2016. Hawkings, J. R., Wadham, J. L., Tranter, M., Lawson, E., Sole, A., Cowton, T., Tedstone, A. J., Bartholomew, I., Nienow, P., Chandler, D., and Telling, J.: The effect of warming climate on nutrient and solute export from the Greenland Ice Sheet, Geochemical Perspectives Letters, 1, 94-104, http://dx.doi.org/10.7185/geochemlet.1510, 2015. Hodson, A., Roberts, T. J., Engvall, A.-C., Holmén, K., and Mumford, P.: Glacier ecosystem response to episodic nitrogen enrichment in Svalbard, European High Arctic, Biogeochemistry, 98, 171-184, 10.1007/s10533-009-9384-y, 2010. Holloway, J. M., and Dahlgren, R. A.: Nitrogen in rock: Occurrences and biogeochemical implications, Global Biogeochem Cy, 16, 10.1029/2002GB001862, 2002. Holmes, R. M., McClelland, J. W., Peterson, B. J., Tank, S. E., Bulygina, E., Eglinton, T. I., Gordeev, V. V., Gurtovaya, T. Y., Raymond, P. A., Repeta, D. J., Staples, R., Striegl, R. G., Zhulidov, A. V., and Zimov, S. A.: Seasonal and Annual Fluxes of Nutrients and Organic Matter from Large Rivers to the Arctic Ocean and Surrounding Seas, Estuar Coast, 35, 369-382, DOI 10.1007/s12237-011-9386-6, 2012. Kalsbeek, F., and Taylor, P. N.: Pb-isotopic studies of proterozoic igneous rocks, west Greenland, with implications on the evolution of the Greenland shield, in: The deep Proterozoic crust in the North Atlantic Provinces, NATO ASI Series, Series C, Mathematical and Physical Sciences, D. Reidal Publishing Company, 1984. Palmer, S., Shepherd, A., Nienow, P., and Joughin, I.: Seasonal speedup of the Greenland Ice Sheet linked to routing of surface water, Earth Planet Sc Lett, 302, 423-428, http://dx.doi.org/10.1016/j.epsl.2010.12.037, 2011. Podgorny, I. A., and Grenfell, T. C.: Absorption of solar energy in a cryoconite hole, Geophys Res Lett, 23, 2465-2468, 10.1029/96GL02229, 1996. Tedesco, M., Fettweis, X., Mote, T., Wahr, J., Alexander, P., Box, J. E., and Wouters, B.: Evidence and analysis of 2012 Greenland records from spaceborne observations, a regional climate model and reanalysis data, Cryosphere, 7, 615-630, DOI 10.5194/tc-7-615-2013, 2013. Tedstone, A. J., Nienow, P. W., Sole, A. J., Mair, D. W. F., Cowton, T. R., Bartholomew, I. D., and King, M. A.: Greenland ice sheet motion insensitive to exceptional meltwater forcing, Proceedings of the National Academy of Sciences, 110, 19719-19724, 10.1073/pnas.1315843110, 2013. Vernon, C. L., Bamber, J. L., Box, J. E., van den Broeke, M. R., Fettweis, X., Hanna, E., and

Huybrechts, P.: Surface mass balance model intercomparison for the Greenland ice sheet, Cryosphere, 7, 599-614, DOI 10.5194/tc-7-599-2013, 2013.